# Different mutant RUNX1 oncoproteins program alternate haematopoietic differentiation trajectories

Sophie G Kellaway* , Peter Keane* , Benjamin Edginton-White , Kakkad Regha, Ella Kennett, Constanze Bonifer

**Mutations of the haematopoietic master regulator RUNX1 are associated with acute myeloid leukaemia, familial platelet disorder and other haematological malignancies whose phenotypes and prognoses depend upon the class of the RUNX1 mutation. The biochemical behaviour of these oncoproteins and their ability to cause unique diseases has been well studied, but the genomic basis of their differential action is unknown. To address this question we compared integrated phenotypic, transcriptomic, and genomic data from cells expressing four types of RUNX1 oncoproteins in an inducible fashion during blood development from embryonic stem cells. We show that each class of mutant RUNX1 deregulates endogenous RUNX1 function by a different mechanism, leading to specific alterations in developmentally controlled transcription factor binding and chromatin programming. The result is distinct perturbations in the trajectories of gene regulatory network changes underlying blood cell development which are consistent with the nature of the final disease phenotype. The development of novel treatments for RUNX1-driven diseases will therefore require individual consideration.**

## Introduction

RUNX1 is a transcription factor which is absolutely essential for haematopoietic development both in vivo and in vitro (Okuda et al, 1996; Lacaud et al, 2002). In humans, different classes of RUNX1 mutations lead to distinct disease phenotypes and clinical outcomes (Bellissimo & Speck, 2017). Mutations involving RUNX1 are one of the most common recurrent drivers of acute myeloid leukaemia (AML) found in around 14% of cases (Papaemmanuil et al, 2016), but also cause other haematological conditions. This includes familial platelet disorder (FPD), acute lymphoblastic leukaemia (Schlegelberger & Heller, 2017), and an association with chronic myelogenous leukaemia (Lugthart et al, 2010). Established leukaemic cells carrying different types of RUNX1 mutations display specific transcriptional and chromatin profiles (Assi et al, 2019). However, in patients, RUNX1 mutations are associated with additional genetic alterations that disrupt differentiation and alter cellular growth (Gaidzik et al, 2016). Therefore, the molecular mechanisms how the sole expression of different types of RUNX1 oncoproteins drive the development of specific disease phenotypes is unclear.

RUNX1 mutations can occur within the DNA-binding domain (DBD), the transactivation domain (TAD), or are a result of translocations resulting in the generation of fusion proteins. RUNX1 functions by directly binding DNA together with its obligate partner CBFβ via the DBD, in large complexes mediated by the TAD (Wotton et al, 1994; Petrovick et al, 1998; Koh et al, 2013). After haematopoietic stem cells have formed, its continued expression during differentiation is not essential but helps pattern and maintain cells in the correct lineage balance (Chen et al, 2009; Cai et al, 2011; Tober et al, 2013), in concert with other transcription factors such as the GATA, C/EBP, and ETS families (Burda et al, 2010; Beck et al, 2013; Goode et al, 2016). Mutations in the DBD are typically point mutations which abrogate binding of RUNX1 to DNA but leave the rest of the protein intact; these are found as germ line mutations in FPD but are also found in AML (Song et al, 1999). Premature stop codons or frameshift mutations typically remove the TAD but may or may not affect the DBD. The latter are typically found in AML with poor prognosis (Mendler et al, 2012; Gaidzik et al, 2016; Döhner et al, 2017) but can also be associated with FPD (Song et al, 1999). Recurrent translocations include t(8; 21), t(3; 21), and t(12; 21), which result in the fusion of part of the RUNX1 protein to all or part of another protein—ETO, EVI1, and ETV6 in the examples given—and are found in AML, chronic myelogenous leukaemia, and acute lymphoblastic leukaemia (Miyoshi et al, 1993; Mitani et al, 1994; Golub et al, 1995; Romana et al, 1995).

The biochemical properties of mutant RUNX1 proteins are well characterised. The functional activities of the different RUNX1 mutations have been studied in detail (Matheny et al, 2007; Ernst et al, 2020; Yokota et al, 2020). DBD-mutated proteins, as expected, cannot bind DNA; they have limited nuclear localisation but maintain CBFβ interaction (Michaud et al, 2002; Matheny et al, 2007).

Institute of Cancer and Genomic Sciences, University of Birmingham, Birmingham, UK

Correspondence: s.g.kellaway@bham.ac.uk; c.bonifer@bham.ac.uk
Regha Kakkad's present address is Department of Ophthalmology, Yong Loo Lin School of Medicine, National University of Singapore, Singapore, Singapore, and Institute of Molecular and Cell Biology (IMCB), Agency for Science, Technology and Research (A*STAR), Singapore, Singapore
*Sophie G Kellaway and Peter Keane contributed equally to this work

TAD mutants can bind DNA with varying efficiency and maintain CBFβ interactions but show very limited nuclear localisation (Michaud et al, 2002; Matheny et al, 2007). Fusion proteins maintaining the RUNX1 DBD are still able to bind DNA, but further interactions are translocation specific, for example, RUNX1-ETO interacts with repressive complexes (Amann et al, 2001). When deleted in haematopoietic stem cells of mice, RUNX1 deficiency causes an increase in immature myeloid cell formation, thrombocytopenia, and lymphocytopenia (Sun & Downing, 2004; Putz et al, 2006). Expression of RUNX1 DBD mutated proteins in mice induces more complex phenotypes, including myelodysplasia and a reduction in colony-forming progenitor cells in the aorta/gonad/mesonephros (Cammenga et al, 2007; Matheny et al, 2007; Watanabe-Okochi et al, 2008). TAD mutant proteins on the other hand, show dosage-dependent phenotypes in mice, with severe disruption to the formation of blood across multiple lineages (Matheny et al, 2007; Watanabe-Okochi et al, 2008). Germ line expression of fusion proteins such as RUNX1-ETO and RUNX1-EVI1 in mice also leads to large scale disruption of blood formation (Okuda et al, 1998; Maki et al, 2005).

It is unclear precisely how the different RUNX1 mutant proteins drive the development of a specific type of disease. Initial hypotheses that these mutations lead to haploinsufficiency of RUNX1 or mediate dominant negative effects do not fully explain disease phenotypes (Cai et al, 2000; Cammenga et al, 2007; Matheny et al, 2007). To address this issue, we carried out a parallel comparative study on two RUNX1 mutants representing DBD and TAD mutations together with two RUNX1 oncofusion proteins (RUNX1-ETO and RUNX1-EVI1) and investigated how they affect transcriptional control and alter RUNX1 driven gene regulatory networks in haematopoietic progenitors. We show that each RUNX1 mutant protein interferes with the RUNX1-driven gene regulatory network in its own way, setting up distinct chromatin landscapes and leading to divergent outcomes of progenitor development.

# Results

## Mutant RUNX1 proteins disrupt haematopoietic differentiation

To understand the individual action of mutant RUNX1 proteins, we utilised a well-characterised embryonic stem cell (ESC) differentiation system, which recapitulates the different steps of haematopoietic specification of blood cells from haemogenic endothelium (HE) and allows inducible expression of oncoproteins (Lancrin et al, 2010; Iacovino et al, 2011; Regha et al, 2015; Goode et al, 2016). We induced each mutant with doxycycline (dox) in otherwise healthy blood progenitor cells (progenitors) at the onset of the RUNX1 transcriptional program (Fig 1A) during the endothelial–haematopoietic transition (EHT) (Lancrin et al, 2010). The RUNX1 mutations studied were R201Q, also reported as R174Q dependent on the RUNX1 isoform, which is a DBD mutant, R204X (also reported as R177X) which is truncated leaving only the DBD. Both mutations have been extensively investigated and the phenotypes they generate were previously studied in transgenic mice (Matheny et al, 2007). To compare these mutants with their fusion protein counterparts, we also studied RUNX1-ETO and

RUNX1-EVI1 (Fig 1A). Induction conditions of each mutant RUNX1 protein were adjusted to ensure that expression levels were near physiological, with cDNA and/or protein expression of the mutant proteins approximately that of the endogenous RUNX1. R204X cDNA was expressed at higher levels than the other cDNAs with the same dox concentration under the same promoter, but this was not reflected by the protein levels (Fig S1A, Regha et al, 2015; Kellaway et al, 2020). As differentiation in this system is not entirely synchronous, the timing of induction was adjusted in a cell line specific manner such that it occurred in approximately the same target cell populations (~30% HE, ~40% progenitors) ensuring that results were comparable between cell lines (Fig S1B).

We first assessed the impact of mutant RUNX1 protein expression on haematopoietic development. We have previously shown that RUNX1-ETO and RUNX1-EVI1 expression impedes the EHT for which RUNX1 is required (Regha et al, 2015; Kellaway et al, 2020), causing a reduced proportion of progenitors and an increased proportion of late HE (HE2) cells, indicating that fusion proteins were impeded the activity of endogenous RUNX1. In contrast, no effect on the EHT was observed with either R201Q or R204X (Figs 1B and S1C). We next investigated how each RUNX1 mutation affected terminal differentiation and self-renewal ability of haematopoietic progenitors. In serial replating assays, we found that the RUNX1 mutants behaved in a mutation-specific fashion (Fig 1C and D). R201Q induction caused an increase in clonogenicity in both primary and secondary colony forming assays. In addition, fewer megakaryocytes formed in the primary colony forming assays (Fig S1D). Expression of R204X and RUNX1-ETO which both cause AML led to an initial reduction in clonogenicity across all lineages, but an increase upon replating, indicative of a differentiation block and enhanced self-renewal. RUNX1-EVI1 expression caused a reduction in both primary and secondary colony forming capacity, again across all lineages, presumably due to the lineage decision promiscuity and cell cycle defects we have previously observed for this protein (Kellaway et al, 2020).

In summary, the four RUNX1 oncoproteins disrupt terminal differentiation in colony forming assays, reflecting the different diseases which they cause, but only the two translocations affected the RUNX1 dependent EHT.

## Endogenous RUNX1 binding changes in response to the presence of oncogenic RUNX1

To investigate the molecular basis of the observed phenotypic differences, we performed RNA-seq, ATAC-seq, and ChIP-seq experiments in c-Kit+ progenitors (Fig S2A) following induction of each of the mutant forms of RUNX1 and integrated the data. We found that changes in chromatin accessibility and gene expression were associated with mutant-specific changes in the endogenous RUNX1-binding patterns (Fig 2). R201Q triggered only minor changes in chromatin accessibility and gene expression after induction but caused a surprisingly large scale reduction in endogenous RUNX1 binding (Figs 2A and S2B) which was reproducibly found in multiple ChIP experiments. This reduction was not caused by direct competition of the R201Q protein with endogenous RUNX1 for chromatin binding, as we were unable to detect binding of the R201Q protein by ChIP, using an antibody against the HA tag (Fig S2C). In contrast,

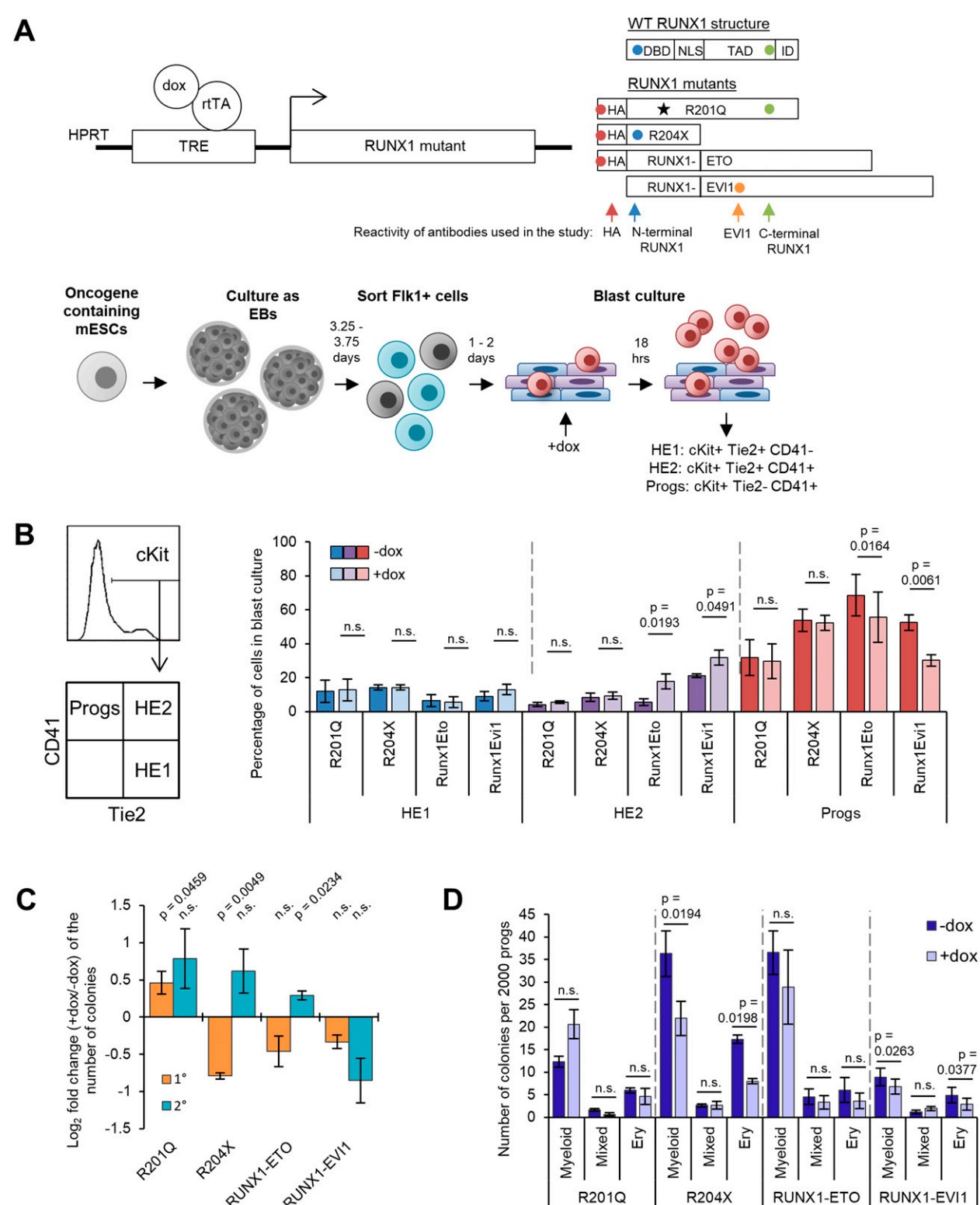

**Figure 1. Induction of RUNX1 mutants during blood differentiation perturbs progenitor identity.**
**(A)** Schematic showing the RUNX1-inducible constructs used, the bindng specificity of the antibodies used in the study, the embryonic stem cell differentiation system, and the stage of induction of the transgenes. **(B)** Flow cytometry was used to assess the proportion of cells in the blast culture which were HE1, HE2, or progenitors as indicated in the schematic on the left. Bars show the mean percentage of cells in each population. N = 3 for R201Q, n = 4 for R204X, and RUNX1-ETO and n = 5 for RUNX1-EVI1.
**(C)** Progenitors were placed into colony forming assays in the without doxycycline. The bars show $\log_2$-fold change of induced (+dox) by noninduced (−dox) for primary colonies in orange, and secondary colonies in blue. R201Q primary colony forming n = 5, n = 3 for all others. **(C, D)** The absolute number of colonies of each lineage subtype

induction of the R204X protein caused little disruption to endogenous RUNX1 binding, but greater changes to chromatin accessibility. The R204X protein again was not found to directly bind chromatin. We cannot exclude the possibility that R201Q and R204X can bind chromatin in a transient fashion, but the signal was below the detection limits of the ChIP experiments. Sites with altered chromatin accessibility after induction of RUNX1-ETO and RUNX1-EVI1 were also seen, with lost sites associated with the loss of endogenous RUNX1, and gained sites associated with gain of RUNX1 binding, with RUNX1-ETO and RUNX1-EVI1 displacing some of the endogenous RUNX1 (Regha et al, 2015; Kellaway et al, 2020). Furthermore, after induction of RUNX1-EVI1, we found an increase in total binding of RUNX1 (Fig S2B). Genome browser screenshots displaying the changes in RUNX1 binding are shown in Figs 2B and S2D; these also highlight that residual RUNX1 binding is preserved at some sites after induction of R201Q but not all.

### RUNX1 binding in the presence of mutant proteins is influenced by altered CBF$\beta$ interactions

Next we questioned whether the changes to endogenous RUNX1 binding were due to the mutant proteins interfering with the interaction of endogenous RUNX1 with CBF$\beta$ using in situ proximity ligation assays (PLAs). By using antibodies specific to either the wild-type RUNX1, HA-tagged–induced mutant proteins, or untagged RUNX1-EVI1, we assessed in single cells whether the induced RUNX1 oncoproteins were complexed with CBF$\beta$ and quantified whether the interaction between CBF$\beta$ and endogenous RUNX1 was affected by oncoprotein induction. We first examined the intracellular localisation of the mutant RUNX1 proteins. Both RUNX1-ETO and RUNX1-EVI1 were clearly localised in nuclei (Figs 3A and S3A, left panels), whereas both R201Q and R204X exhibited diffuse staining with little protein found in the nucleus, consistent with previous studies (Osato et al, 1999; Michaud et al, 2002). We then examined whether induced proteins interacted with CBF$\beta$ and where in the cell. We found a high number of interactions between RUNX1-ETO (measured using the HA antibody) and CBF$\beta$, and RUNX1-EVI1 (measured using the EVI1 antibody) and CBF$\beta$ located within the nucleus (Figs 3A and S3A, right panels). In contrast, we observed very few interactions between R201Q (HA antibody) and CBF$\beta$, or R204X (HA antibody) and CBF$\beta$ compared with background. Interestingly, despite minimal nuclear localised R201Q and R204X protein, we saw PLA foci in the nucleus, suggesting that some mutant RUNX1-containing complexes were capable of nuclear translocation.

We next assessed the quantity of interactions of the endogenous RUNX1 with CBF$\beta$ and compared them with the ChIP-seq results. Using antibodies against wild type RUNX1 and CBF$\beta$ alone showed the expected staining patterns which were unaffected by dox induction (Fig S3B). In the uninduced cells, the number of PLA foci was similar for all cell lines allowing us to see only the effects of the mutant proteins (Fig 3B, $P$-value = 0.723 by one-way ANOVA). RUNX1-ETO and R204X expression caused no change in the number of

RUNX1/CBF$\beta$ interactions; R204X induction did not affect RUNX1 binding in chromatin, disruption of RUNX1 binding caused by induction of RUNX1-ETO was therefore predominately due to displacement of RUNX1 by RUNX1-ETO, as previously shown. RUNX1-EVI1 expression caused an increase in the number of RUNX1/CBF$\beta$ foci (Figs 3A and S3A, centre panels) which mirrored the ChIP-seq data where we saw increased RUNX1 binding (Fig 2). Most strikingly, however, given the mild phenotype, R201Q expression caused a reduction in the number of PLA foci, explaining the decrease in the amount of RUNX1 available to efficiently bind chromatin (Fig 2). The RUNX1 antibody used for this assay was unable to discriminate the endogenous RUNX1 from the induced R201Q, and therefore some of these foci may in fact be R201Q/CBF$\beta$ interactions, meaning RUNX1/CBF$\beta$ interactions were even further reduced than measured.

Taken together, these data show that binding of the endogenous RUNX1 is disrupted by the expression of mutant RUNX1 proteins, with mutation-specific changes in the frequency of interactions between endogenous RUNX1 and CBF$\beta$. We found no evidence that CBF$\beta$ was stably sequestered by mutant RUNX1 proteins, although it is possible that CBF$\beta$ is sequestered and then degraded. Direct displacement of endogenous RUNX1 chromatin binding by the mutant RUNX1 proteins was only found in the case of the two fusion proteins. Taken together, these experiments demonstrate that the expression of mutant RUNX1 impacts on RUNX1 chromatin binding and the chromatin landscape in a mutation class-specific fashion.

### RUNX1 binding alterations lead to mutation class-specific changes in gene regulation

To understand how the disruption of the RUNX1 developmental program drives the observed phenotypes and to examine whether the mutant RUNX1 forms target similar transcriptional networks, we compared gene expression changes using RNA-Seq. Overall gene expression patterns for induced and uninduced cells were highly consistent across the four cell lines (Fig S4A) and replicates correlated well (Fig S4B). As expected from the cell biological data, RUNX1-ETO and RUNX1-EVI1 de-regulated the most genes across the EHT, and similarly, fewer changes were seen following induction of R201Q and R204X (Figs 4A and S4C and Table S1). After induction of R201Q, the vast majority of genes continued to be regulated according to their expected trajectory, with a subset failing to be up-regulated to the extent they normally would, including *Hba-a1*, *Cd79b*, and *Mef2c*. The induction of RUNX1-ETO caused the greatest number of genes to not be down-regulated sufficiently, including *Gfi1* ([Lancrin et al, 2012], Fig 4A and Table S1). Looking specifically at the changes at the specific cell stages, RUNX1-ETO and RUNX1-EVI1 induction both caused the greatest number of genes to be up- or down-regulated in both HE2 and progenitors, and R204X induction only caused up-regulation of genes at the HE2 stage and not in progenitors, for example, *Mecom* and *Plek* (Fig S4C).

We then examined whether the mutant RUNX1 proteins were targeting the same transcriptional networks. We first performed

---

from the primary colony forming assays in (C) is shown. Data information: (B, C, D) error bars show standard error of the mean. *P*-values were calculated using paired *t* tests between–and +dox pairs, n.s. indicates *P* > 0.05.

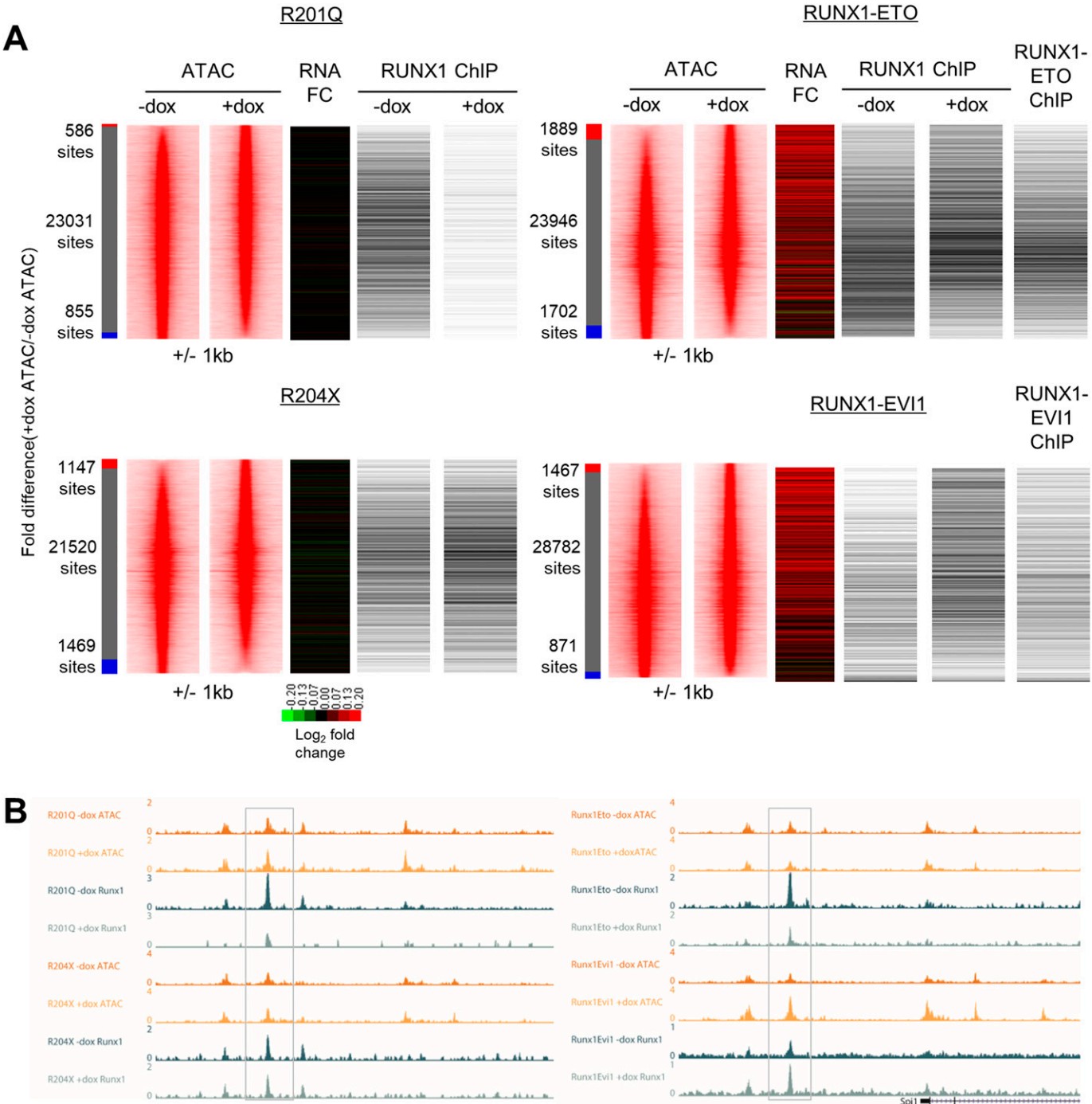

**Figure 2. Mutant RUNX1 induction leads to specific changes in endogenous RUNX1 binding and chromatin accessibility.**
**(A)** Chromatin accessibility in cKit+CD41+Tie2–sorted progenitors at distal sites as determined by ATAC-seq was ranked by fold change of the +dox/−dox tag count and represented as density plots (±1 kb from the summit). The gene expression fold change as determined by RNA-seq (+dox/−dox) was plotted alongside based on nearest gene assigned. The binary presence or absence of a RUNX1, RUNX1-ETO, or RUNX1-EVI1 ChIP peak was also plotted based on intersection with the open chromatin. The red bar indicates +dox-specific sites, grey shared and blue −dox-specific sites where the normalised tag-count of specific sites was at least twofold different. **(B)** University of California Santa Cruz (UCSC) Genome browser screenshot of counts-per-million-normalised ATAC-seq and ChIP-seq tracks at the *Spi1* locus. The box highlights the *Spi1* enhancer which demonstrates changes in RUNX1 binding and chromatin accessibility.

pair-wise analysis, to see whether different mutant proteins cause similar (Fig 4B, left–red indicates mutually up-regulated, blue indicates mutually down-regulated) or opposing (Fig 4B, right–horizontal/blue indicates down-regulated while vertical/red indicates up-regulated)

changes in gene expression patterns. This analysis showed that just under a quarter of the genes which were up-regulated in HE2 after R204X induction were also up-regulated in HE2 after RUNX1-EVI1 induction (Fig 4B, left); a greater number of these genes were

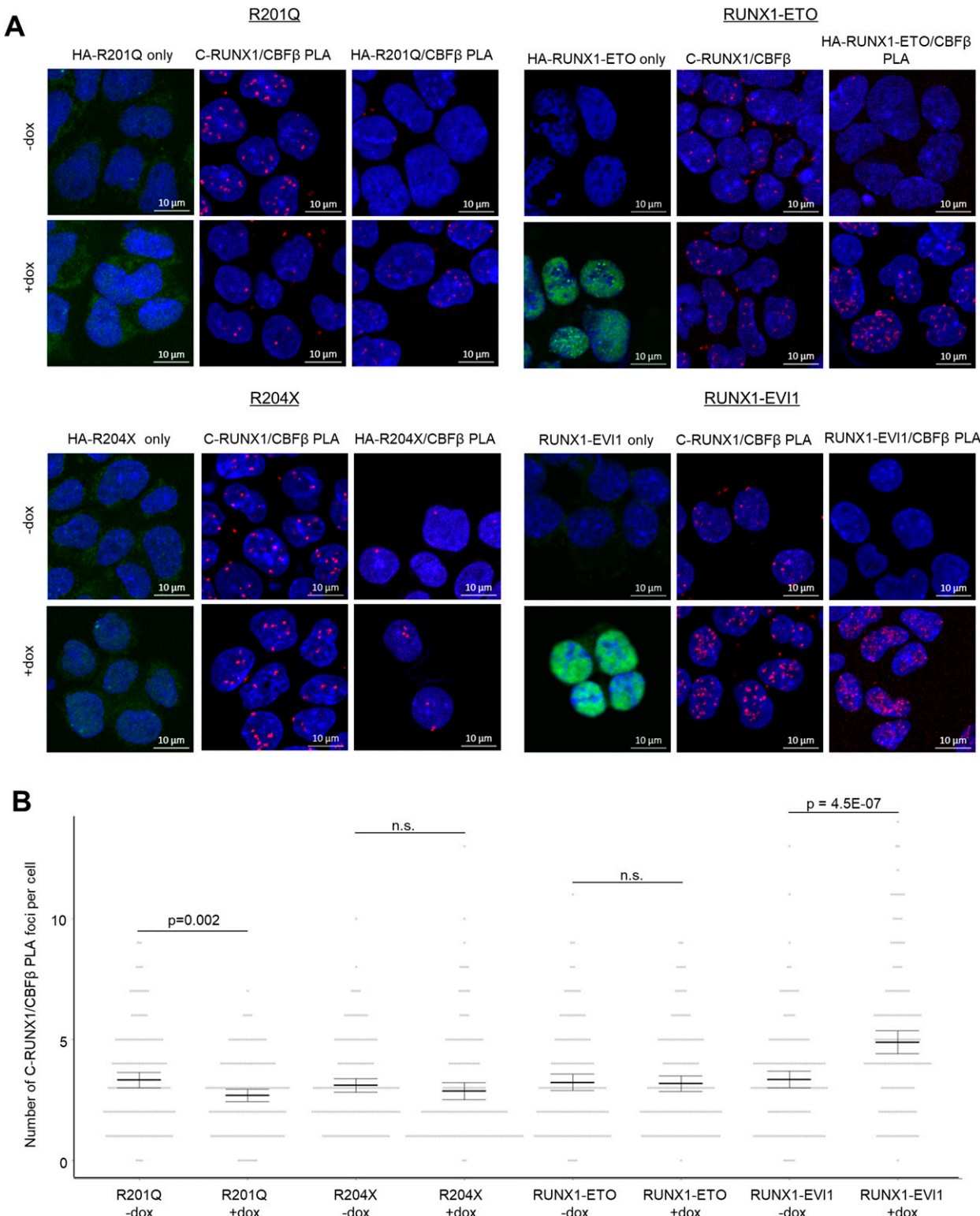

**Figure 3. RUNX1 mutants interact with CBFβ and partially disrupt RUNX1/CBFβ interactions.**
**(A)** Representative images are shown of immunocytochemistry and proximity ligation assay (PLA) in progenitors with and without induction of the mutant forms of RUNX1. For each cell line, the left panel shows immunocytochemistry of the mutant protein alone (shown in green, using anti-HA or anti-EVI1 antibodies) counterstained with DAPI (blue). The centre panel shows a PLA of endogenous RUNX1 (cross reaction with R201Q possible) using C-terminal RUNX1 antibody with CBFβ (red), with DAPI (blue). The right panel shows a PLA of the mutant RUNX1 with CBFβ (red), with DAPI (blue). **(B)** The number of endogenous RUNX1/CBFβ PLA foci were counted in 150 cells across three biological replicates and are shown by the grey circles. The mean and 95% confidence intervals are indicated by the bar and error bar. *P*-values were calculated using two-sample *t* tests between−and + dox pairs, n.s. indicates a *P*-value > 0.05.

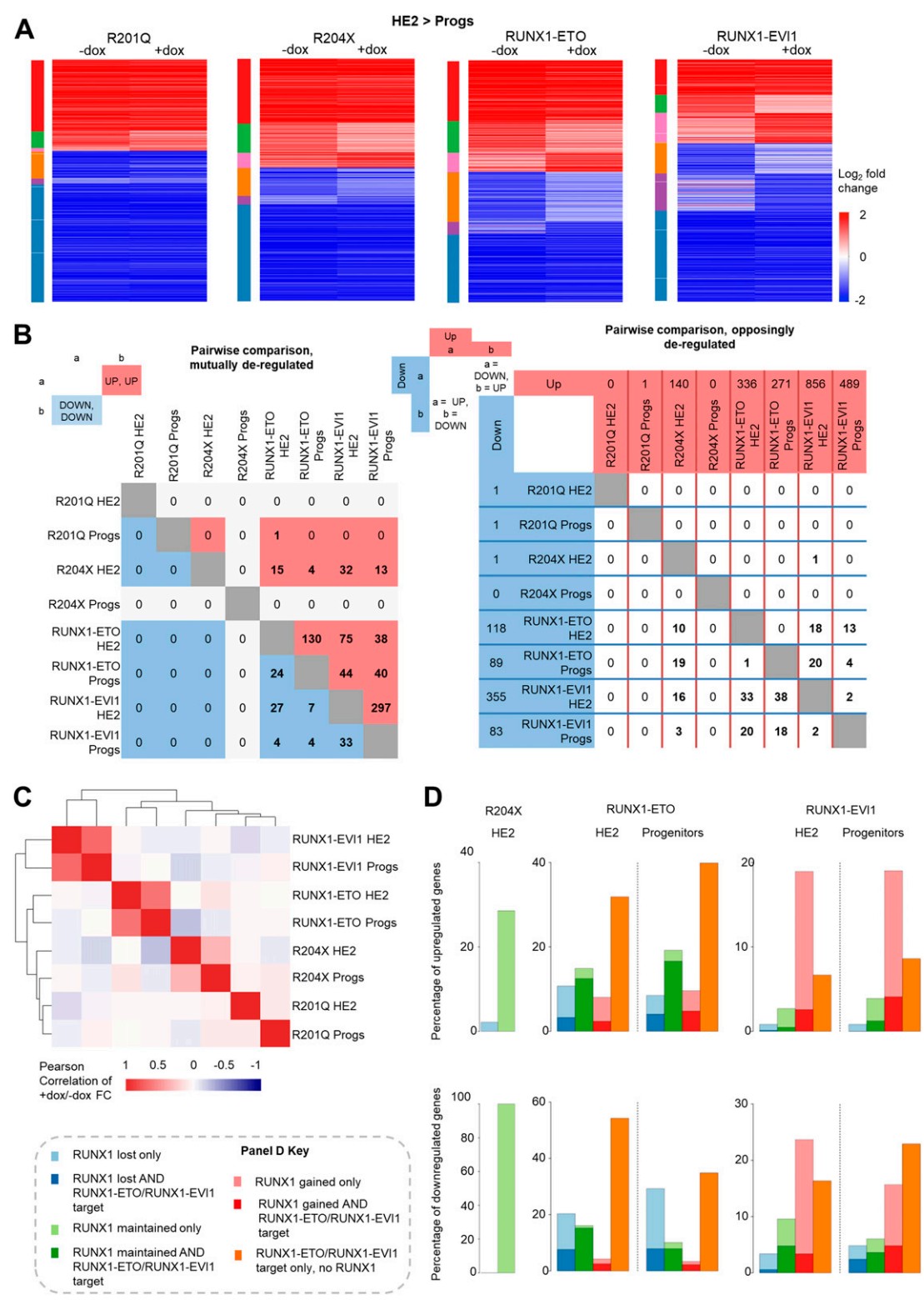

**Figure 4. Mutant forms of RUNX1 cause unique and shared gene expression changes.**
**(A)** Heat maps showing the log$_2$-fold gene expression changes across the HE2 to progenitor transition. Colour bars on the left indicate genes which are (red) up-regulated in both – and +dox (green), up-regulated in –dox only, (pink) up-regulated in +dox only (orange) down-regulated in –dox only, (purple) down-regulated in +dox only (blue), down-regulated in both – and +dox. **(B)** Pairwise analysis of genes which were twofold up- or down-regulated in either HE2 or progenitors after induction of each RUNX1 mutant. The left table shows the number of genes which were mutually up (red) or down-regulated (blue), the right table shows the number of genes which were up-regulated in the dataset shown along the top and down-regulated in the dataset on the side. Columns or rows which are greyed out have 0 genes deregulated in

down-regulated in progenitors after RUNX1-ETO induction rather than up-regulated (Fig 4B, right), indicating complex stage-specific regulation. In a similar vein, multiple genes which were up-regulated after RUNX1-EVI1 induction were both up- or down-regulated after RUNX1-ETO induction, in both HE2 and progenitors. Furthermore, genes which were down-regulated after RUNX1-EVI1 induction were predominately up-regulated after RUNX1-ETO induction, particularly in progenitors, indicating opposing regulatory mechanisms.

We next performed correlation analysis and hierarchical clustering based on all genes whose expression was changing across all of the datasets (Fig 4C). This analysis showed that genes affected by RUNX1-EVI1 were mostly unique, and to a lesser extent this was true for all other mutant RUNX1 driven gene expression changes indicating mutant-specific genomic impacts. However, this analysis also indicated an inverse correlation in gene expression changes caused by R204X and RUNX1-ETO which was of note as these both drive AML and contain the RUNT-domain portion of RUNX1.

Next, we analysed which of the genes with altered expression were direct targets of either RUNX1 or the two fusion proteins, based on our ChIP-seq data (Fig 2) and examined whether RUNX1 binding was lost, maintained or gained in response to induction of the mutant RUNX1 proteins (Fig 4D). None of the three genes deregulated by R201Q were RUNX targets. RUNX1-EVI1 binding caused genes to be down-regulated but most changes in gene expression were driven by the large scale increase in RUNX1 binding (Figs 4D and 2), whereas loss of RUNX1 and RUNX1-ETO binding to these sites correlated with gene expression changes seen in response to RUNX1-ETO induction. A large proportion of the genes up-regulated in response to R204X induction were RUNX1 targets but binding of RUNX1 was unchanged, again indicating that this oncoprotein perturbs the action of RUNX1 at its binding sites rather than disrupting binding itself.

The impact of the RUNX1 oncoproteins on gene expression was, therefore mild, varied and generally occurred in a mutation specific fashion, despite a significant proportion of affected genes being RUNX1 targets.

### Mutant oncoproteins disrupt RUNX1-mediated transcription factor and chromatin reorganisation

We previously showed that the up-regulation of RUNX1 during haematopoietic specification leads to a global reorganisation of transcription factors binding and chromatin patterns (Lichtinger et al, 2012; Gilmour et al, 2018). We therefore hypothesised that the RUNX1 mutants may interfere with this process and disrupt the transcription factor hubs that provide instructions for further blood cell differentiation. We first examined the transcription factor–binding motifs associated with differential chromatin accessibility and found that the patterns of motif enrichment were specific to each RUNX1 mutant (Figs 5A and S5A). With R201Q we found an increase in chromatin accessibility associated with GATA motifs,

after RUNX1-ETO induction accessible sites associated with RUNX and PU.1 were lost, and after RUNX1-EVI1 induction sites containing GATA and RUNX motifs were lost, but PU.1 sites were gained. Interestingly, following induction of R204X–which lacks a trans-activation domain–accessible chromatin sites were both lost and gained (Fig 5B) but were not associated with any changes in motif enrichment. RUNX motifs were also unchanged after R204X expression suggesting it is not acting as dominant negative to the endogenous RUNX1 which again echoes the phenotypic and ChIP-seq data.

We confirmed the cause of two of the changes in motif composition by performing ChIP-seq for the transcription factors which bind to them. With R201Q, increased accessibility at GATA motifs was associated with overall increased GATA1 (the GATA factor most highly expressed in progenitors) binding, whereas in RUNX1-EVI1 expressing cells, PU.1 binding was most prevalent at those sites where chromatin accessibility was gained (Fig 5C) but was reduced overall (Fig S5B). These results highlight a profound disturbance of RUNX1-driven transcription factor binding reorganisation.

Alongside the changes associated with transcription factor binding we investigated whether lost or gained ATAC-seq peaks were shared or specific for each RUNX1 mutant. All –dox samples were generally well correlated allowing a comparison between changes caused by each oncoprotein (Fig S5C). We calculated the union of all differential peaks and ranked them in parallel, ordered by the R201Q –dox sample (Fig 5D). As with the RNA-seq experiments (Fig 4C), this analysis again showed that each mutant RUNX1 altered the accessible chromatin landscape in a specific fashion with only a few common differentially accessible regions. We noted an inverse pattern of alterations caused by R204X and RUNX1-ETO induction. To further examine this finding, we performed a correlation analysis using the tag counts for each sample across all differentially accessible peaks (Fig 5E), which again showed that R204X +dox and RUNX1-ETO –dox and R204X –dox and RUNX1-ETO +dox each cluster together although the majority of differentially accessible peaks were still unique (Fig S5D). Taken together with the RNA-seq data, these results suggest that R204X and RUNX1-ETO induction affects similar gene regulatory networks which may be why they cause a similar phenotypic outcome.

RUNX1 associated transcription factor complexes can include histone acetyltransferases, which RUNX1-ETO in particular is known to disrupt (Wang et al, 1998; Amann et al, 2001). We therefore examined whether RUNX1-mutant specific chromatin changes were associated with altered histone acetylation patterns. H3K27ac patterns were globally affected by R204X, RUNX1-ETO, and RUNX1-EVI1 induction, with acetylation both lost and gained around accessible chromatin (Fig 6A).

Lost or gained histone acetylation was not exclusively linked to lost or gained chromatin accessibility. The small number of chromatin changes observed in response to R201Q or R204X expression were reflected in the H3K27ac alterations at these sites (Figs 6B and S6A), but levels of H3K27ac strongly increased or

one of the datasets therefore cannot have any in common. **(C)** Heat map showing the Pearson correlation with hierarchical clustering of the +dox/−dox fold change for all deregulated genes across all eight datasets. **(D)** The percentage of up or down-regulated genes associated with RUNX1 and/or RUNX1-ETO or RUNX1-EVI1 ChIP peaks is plotted.

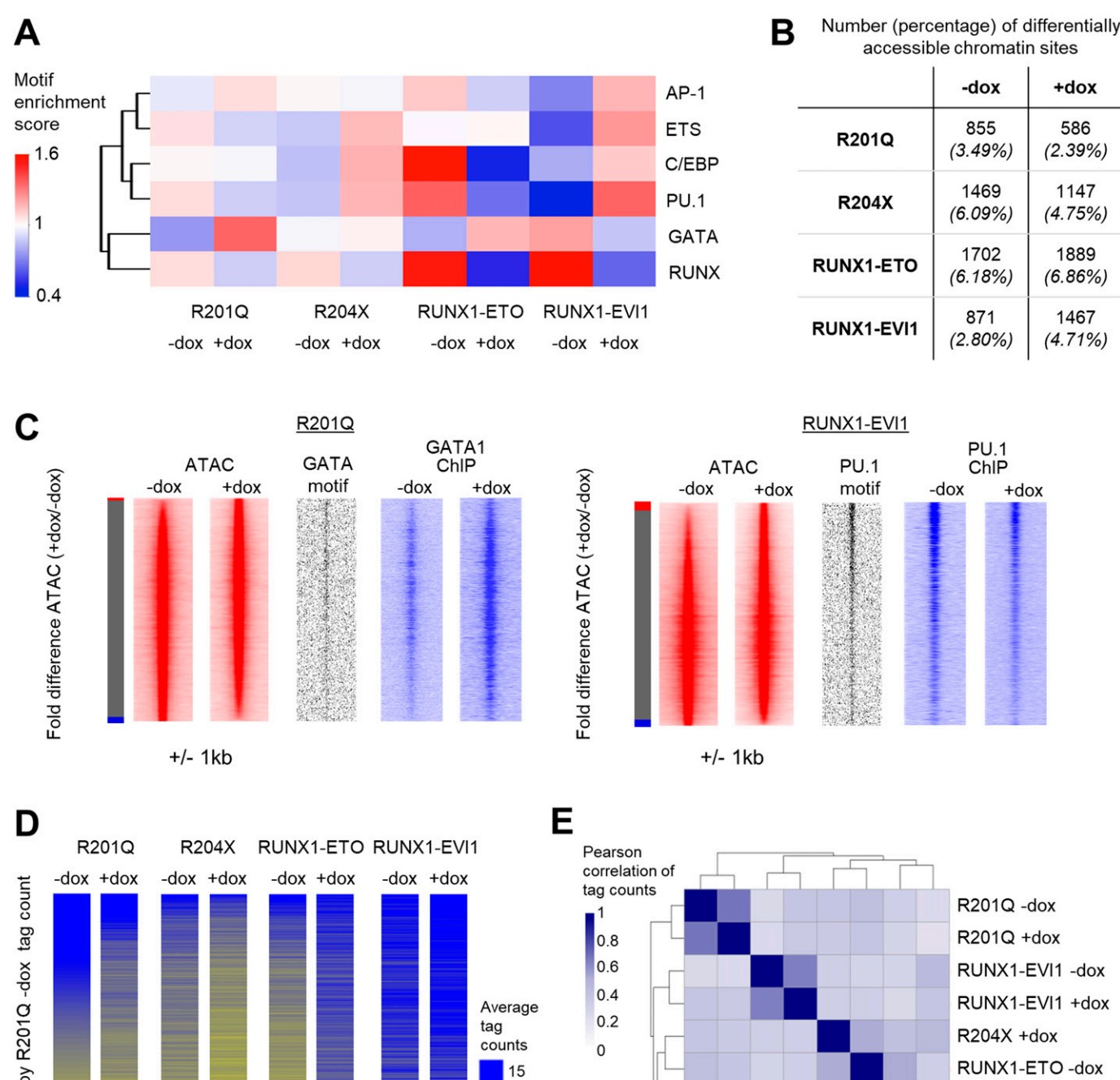

**Figure 5. Chromatin accessibility changes are unique to each RUNX1 mutant and correlate with specific transcription factor–binding patterns.**
**(A)** Heat map of hierarchical clustering, showing the normalised enrichment score for transcription factor motifs which were seen in the de novo motif search of specific distal ATAC sites in progenitors. **(B)** Table showing the number of specific ATAC peaks in progenitors, and the percentage of the total peaks this corresponds to. **(C)** Chromatin accessibility in progenitors was ranked by fold change of the +dox/−dox tag count and represented as density plots (±1 kb), as depicted in Fig 2A. Motif enrichment and ChIP-seq of key transcription factors are plotted alongside. **(D)** ATAC tag counts were calculated across union of all −/+dox-specific distal peaks across all four RUNX1 inductions in progenitors, and ranked according to R201Q −dox descending tag count. 9,494 unique peaks of 9,986 used were unique. **(B, E)** Heat map showing the Pearson correlation and hierarchical clustering using the tag counts of the union of specific peaks in progenitors calculated in (B).

decreased at sites with similarly altered chromatin accessibility after RUNX1-ETO and RUNX1-EVI1 binding, coinciding with up or down-regulation of the associated genes (Fig 2A). Interestingly, at accessible chromatin sites lost after RUNX1-EVI1 expression (Fig 6B), we observed a pattern consistent with flanking histones moving together which is indicative of a loss of transcription factor complexes at these sites (Bevington et al, 2017).

As expected from the initial analysis, differential H3K27ac sites were not associated with differential chromatin accessibility except in the case of RUNX1-ETO (Figs 6C and S6B). Furthermore, differential H3K27ac sites were only minimally linked to altered RUNX1 binding, again predominately following RUNX1-ETO induction, suggesting that although these changes are occurring at sites of RUNX1 binding, they result from perturbation of a larger complex. We also noted that a greater proportion of the sites which lost H3K27ac after induction of R204X, RUNX1-ETO, or RUNX1-EVI1 were shared as compared with the sites which gained H3K27ac (Fig 6D), which may indicate shared sites of repression by these proteins.

Collectively, our data show that in spite of the relatively modest changes in gene expression after induction of some of the RUNX1 oncoproteins, the induction of all of them interferes with RUNX1 activity and rapidly alters transcription factor occupancy and histone modification patterns.

### Mutant RUNX1 proteins alter lineage-specific chromatin priming

The developmentally controlled activation of differential gene expression programs during haematopoietic specification requires the gradual reorganisation of chromatin often preceding the onset of tissue specific gene expression, known as chromatin priming (Goode et al, 2016; Bonifer & Cockerill, 2017). Because RUNX1 is essential for the establishment of a blood cell–specific chromatin landscape (Lichtinger et al, 2012), we hypothesised that despite causing limited alterations in gene expression, each mutant RUNX1 protein may uniquely perturb the chromatin architecture and transcription factor regulatory networks to differentially prime haematopoietic progenitor cells and thus derail future development.

We therefore analysed the degree to which the previously identified differentially accessible chromatin sites (Fig 2A) were shared with different precursor and mature cell types. The ATAC-seq data used for this analysis were derived from purified common myeloid progenitor (CMP), B cell, monocyte, erythroblast, and megakaryocyte to cover the key lineage branches which RUNX1 mutation is known to influence (Lara-Astiaso et al, 2014; Heuston et al, 2018). A cell type–specific chromatin signature was calculated for each cell type by identifying only those peaks which were not shared between different cell types. This set of peaks was compared with the differentially accessible chromatin sites formed or lost after induction of RUNX1-mutant proteins, as shown in the schematic in Fig 7A. An enrichment (Z) score was determined by comparing them to randomly sampled peaks within the union of all accessible sites for all cell types (values shown in Source Data for Fig 7B). In a healthy progenitor cell, we would expect to see balanced lineage priming for mature cells, as well as the progenitor cell signature. By examining the specifically lost or gained sites, we

could therefore understand how the RUNX1 mutants perturbed lineage priming.

After R201Q induction, we found that lost accessible chromatin sites (Fig 7B, lower panel) were highly enriched for a megakaryocyte signature, whereas those sites which were gained showed a slight enrichment for the erythroid fate (Fig 7B, upper panel), indicative of a skew in the megakaryocyte-erythroid branch of blood cell development (Figs 7B and S7). Both lost and gained sites also showed enrichment for B cell–primed sites, although to a lesser degree in the gained sites. None of the sites which gained chromatin accessibility were associated specifically with the monocyte lineage. A different pattern of changes was seen with R204X induction, where megakaryocyte priming was strongly enriched in sites where chromatin accessibility was gained but also slightly enriched in lost sites as well, again suggesting a disruption of differentiation rather than a clear change of cell fate. Priming for all other lineages was preserved, but we found a significant absence of sites associated with CMPs in sites which lost accessibility after expression of R204X suggesting a preservation of the CMP-specific chromatin state.

Both fusion oncoproteins caused a greater disruption in the balance of lineage priming, in line with them causing increased phenotypic and gene expression changes. RUNX1-ETO expessing cells gained accessibility at sites associated with both CMPs and the B-cell lineage, an example of which is shown in Fig S7, and with a reciprocal lack of these lineages losing chromatin accessibility. At the same time, sites specific for the megakaryocyte lineage were lost, and a small proportion of sites associated with the monocytic lineage were gained. Similarly, RUNX1-EVI1 expression caused widespread disruption of priming but with no one lineage pattern specifically gained or lost. Here, sites associated with B cells and megakaryocytes were gained (which can also be seen in Fig S7) and sites associated with monocytes and erythroblasts were lost. Concordantly, erythroblast lineage-specific chromatin sites were not gained in response to RUNX1-EVI1, nor were CMP sites. In summary, our data show that RUNX1-mutant proteins each influence the RUNX1-driven reorganisation of chromatin accessibility and lineage priming in unique ways leading to a disturbance of differentiation trajectories.

## Discussion

In this study, we show that different mutations in *Runx1* give rise to proteins which uniquely disrupt the gene regulatory networks at the onset of blood cell differentiation. During the EHT, RUNX1 reorganises the epigenetic and transcription factor–binding landscape to repress the endothelial fate and primes chromatin for continued haematopoietic differentiation (Lie-A-Ling et al, 2014; Gilmour et al, 2018). Chromatin priming at this stage by RUNX1 binding and elevated histone acetylation is critical for the correct binding patterns of transcription factors driving differentiation, such as PU.1 (Creyghton et al, 2010; Lichtinger et al, 2012). Our study uncovered a profound impact on lineage-specific chromatin priming as a result of perturbation of RUNX1 function at this stage which is then reflected in the composition of terminally differentiated cells. Most importantly, for RUNX1 point mutants, this

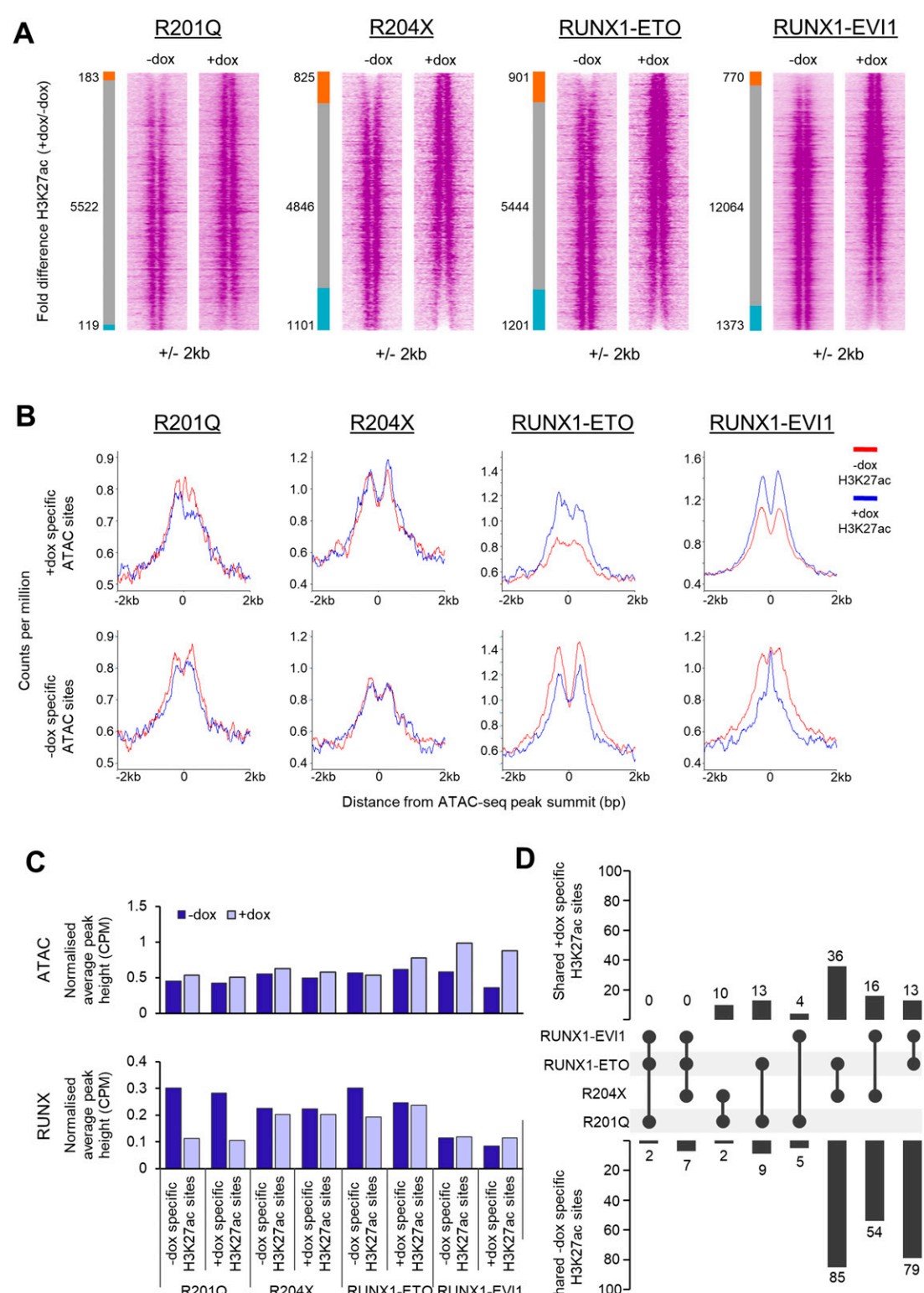

**Figure 6. H3K27ac changes caused by RUNX1 mutants are not wholly dependent on changing chromatin accessibility.**
**(A)** The H3K27ac ChIP-seq signal at open chromatin sites in progenitors was ranked by fold change of the +dox/−dox tag count and represented as density plots (±2 kb). The side bar indicates +dox-specific sites (orange), grey shared, and blue −dox-specific sites where specific sites are at least twofold different. The number of sites shared, lost or gained is indicated.
**(B)** Average profiles of H3K27ac counts-per-million-normalised ChIP-seq signal in progenitors plotted around the differential distal ATAC sites identified in Fig 5 (±2 kb).
**(B, C)** The counts-per-million-normalised average peak heights of ATAC-seq and RUNX1 ChIP-seq were calculated for the specific sites identified in (B). **(B, D)** The percentage of shared specific sites identified in (B) was calculated and shown by the bar graphs, where the circles indicate sets which have been overlapped in each case. Sets where there are no intersecting sites in either the − or +dox-specific sites are not shown.

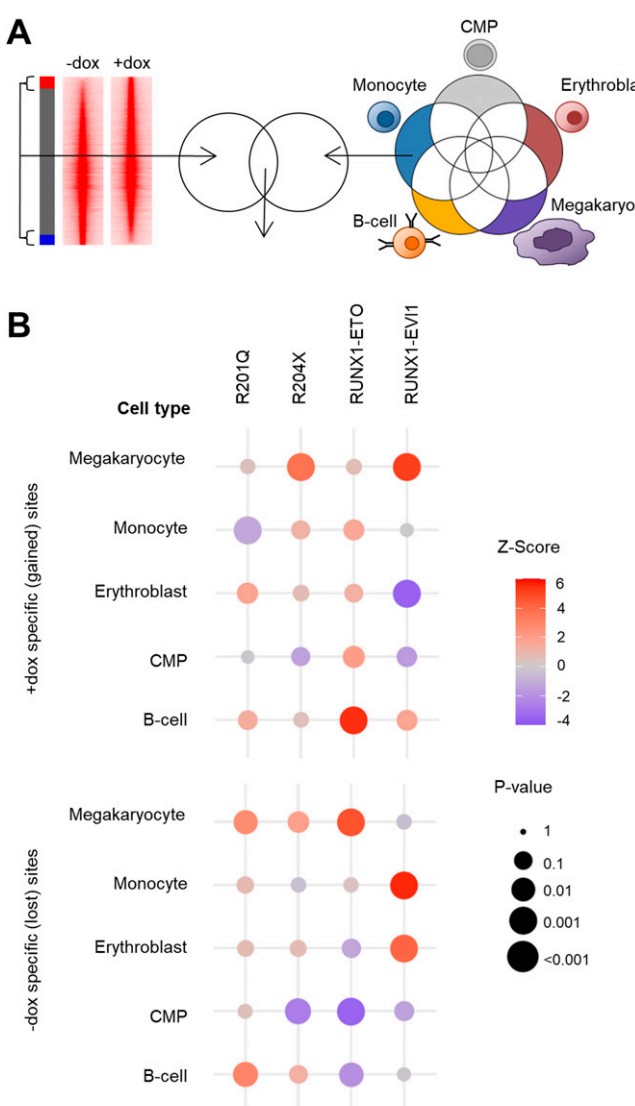

**Figure 7. RUNX1 mutants disrupt RUNX1-driven chromatin priming.**
**(A)** Scheme of how the enrichment of differentially accessible ATAC-seq peaks from Fig 2A, intersecting with ATAC peaks specific to common myeloid progenitors, B-cells, monocytes, erythroblasts or megakaryocytes was calculated. **(B)** Bubble plots showing the association of differentially accessible peaks after mutant RUNX1 induction with each peak set from the indicated cell types. Each bubble represents one intersection, the Z-score representing level of enrichment (red) or depletion of sites of each lineage (blue) as shown by the colour scale. The *P*-value is shown by the size of the circle.
Source data are available for this figure.

perturbation occurred with only minimal influence on gene expression. RUNX1-ETO affected chromatin accessibility associated with RUNX, PU.1, and C/EBP motifs, leading to a skew in progenitor/myeloid differentiation, which was mirrored in an increased immaturity of colony forming cells. We also found a bias towards the B-cell lineage, as it has been previously implicated in t(8; 21) leukaemia (Pabst et al, 2001; Tagoh et al, 2006; Ray et al, 2013; Sun et al, 2013). Similar defects in lineage priming and phenotypic outcomes were seen for R204X, with increased stemness as the primary consequence. RUNX1-EVI1 affected the megakaryocyte/erythroid

balance according to our lineage priming analysis, as well as disrupting the monocyte lineage, and this feature was reflected in a reduced differentiation to myeloid and erythroid cells. Conversely, R201Q caused defects in megakaryopoiesis in colony-forming cell assays, with a concordant gain of GATA1 binding. It has been previously hypothesised that impaired erythropoiesis caused by RUNX1-DBD mutants was due to a change in RUNX1/GATA1 balance at the onset of erythroid differentiation (Waltzer et al, 2003; Cammenga et al, 2007). Our global binding data confirm this idea. RUNX1 is normally required to block the erythroid fate in favour of the megakaryocyte fate (Song et al, 1999; Kuvardina et al, 2015). Megakaryocytic differentiation is, therefore, dependent on the RUNX1/GATA1 balance as well (Elagib et al, 2003), suggesting a likely mechanism by which these RUNX1-DBD mutants contribute to platelet disorders.

One outstanding question has been the degree to which RUNX1-mutant phenotypes result from *RUNX1* haploinsufficiency due to the mutant proteins being nonfunctional or acting in a dominant negative fashion. Previous studies expressing mutant RUNX1 proteins in mice have shown them to have weakly dominant negative or null activity, as blood cell formation was inhibited (Matheny et al, 2007). Some aspects of the mechanism by which this phenotype develops were inferred by studies that inhibited of RUNX1-controlled myeloid gene expression (Guo et al, 2012). However, because of the strong disruption in blood cell formation, the earliest events of cellular reprogramming by mutant RUNX1 proteins could not be studied. By inducibly expressing the mutant proteins on a background of wild-type RUNX1, we demonstrate that all mutant proteins have additional functions.

From our data, we developed a model of how the different mutant RUNX1 proteins interfere with wild-type RUNX1 to disrupt the control of differentiation (Fig 8). Expression of the R201Q (DBD mutant) leads to a reduced interaction of wild-type RUNX1 with CBFβ, a drastic reduction of global RUNX1 binding, increased GATA1 binding and thus a bias away from megakaryocyte differentiation. R204X (which lacks its TAD) does not affect the binding of wild-type RUNX1 or other transcription factors but instead leads to changes in histone acetylation affecting the CMP trajectory. RUNX1-ETO displaces wild-type RUNX1, leads to reduced expression and binding of PU.1 and C/EBPα (Pabst et al, 2001), and to reduced histone acetylation—thus blocking cell differentiation at the early multipotent precursor cell stage and priming these cells towards a B-cell identity. RUNX1-EVI1 acts in a similar fashion to RUNX1-ETO but in addition causes increased RUNX1 binding associated with increased CBFβ interaction which has a knock-on effect on transcription factors such as PU.1 causing widespread disruption of all lineages in which RUNX1 is involved.

In summary, our study has elucidated the genome-wide changes caused by four mutant RUNX1 proteins and shown that they disrupt the earliest instructions for the differentiation trajectory of haematopoietic progenitors. Full-length RUNX1 is required to rescue haematopoiesis in *RUNX1* knockout embryos and to set up balanced haematopoiesis (Goyama et al, 2004), which will require the establishment of correct lineage priming at the chromatin level. The expression of RUNX1-mutant proteins that lack different domains of the protein disturbs this process. The germ line expression of RUNX1-ETO and RUNX1-EVI1 is incompatible with normal blood cell development (Yergeau et al, 1997; Maki et al, 2005). However, RUNX1 point mutations can run in families (Song et al, 1999) and permit haematopoiesis which is in line with the results shown here.

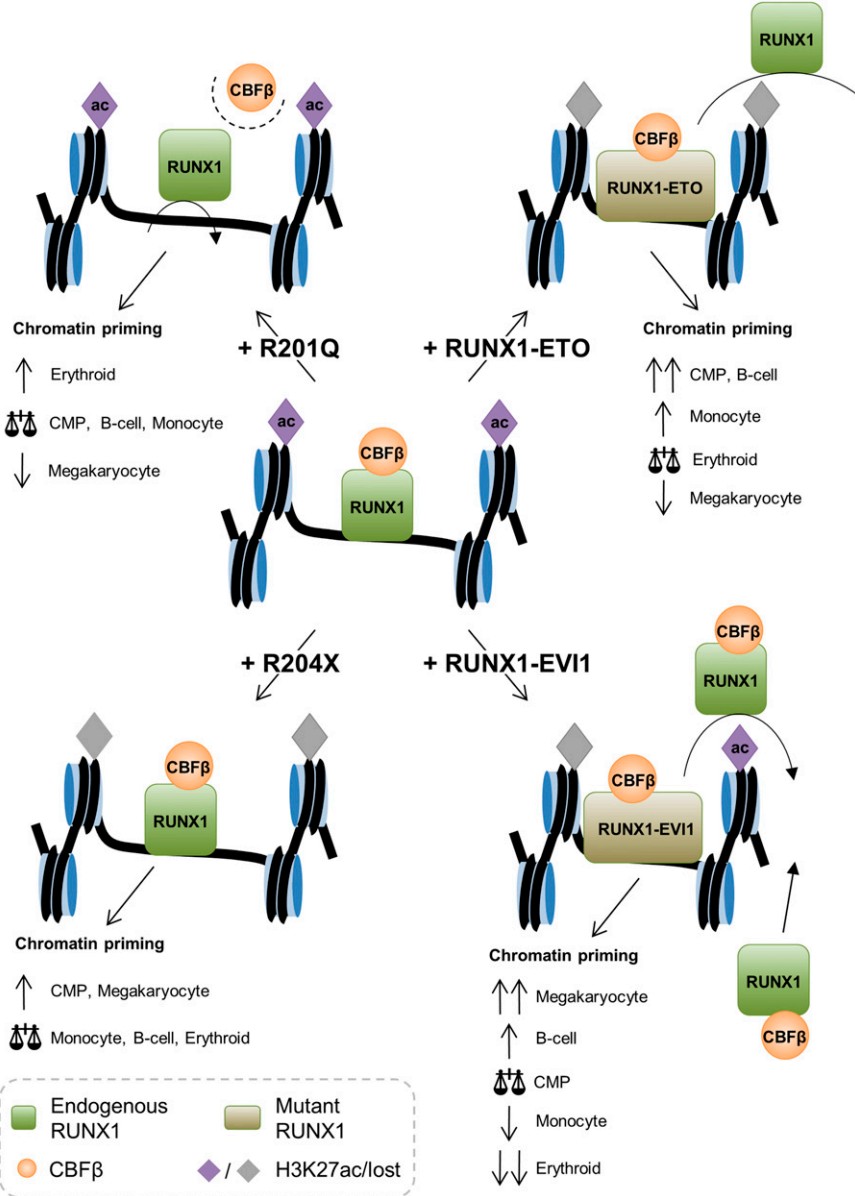

**Figure 8. Mechanism of RUNX1 oncoprotein action on chromatin priming.**
A model for how each of the RUNX1 mutants is disrupting the normal activity of wild-type RUNX1 (centre) based on the data we have generated.

Our data would, therefore, predict that affected individuals in such families would already display signs of deregulation in the chromatin patterns of their progenitor cells. Our results demonstrate that different classes of mutation in RUNX1 have unique multifactorial mechanisms of contributing to disease and so development of novel treatments will require an individual approach.

# Materials and Methods

### Mouse RUNX1-EVI1 ESC generation

Generation of RUNX1-ETO and RUNX1-EVI1–containing ESCs was previously described (Regha et al, 2015; Kellaway et al, 2020). R201Q and R204X plasmids were generated by site-directed mutagenesis on wild-type human *RUNX1c*, and N-terminal HA tags added using the following primers: R201Q forward 5′-CAGTGGATGGGCCCCAA-GAACCTCGAAGAC-3′, reverse 5′-GTCTTCGAGGTTCTTGGGGCCCATC-CACTG-3′ and R204X forward 5′-CCCCCTCGAGCCACCATG-3′, reverse 5′-GCCGATGATATCTCAAGGTTCTCG-3′. A2lox ESCs (a gift from Michael Kyba) were transduced with 20 µg of each plasmid using the 4D-Nucleofector (Lonza), mouse ES program and P3 primary cell kit. Note, R201Q, R204X, and RUNX1-ETO all included N-terminal HA-tags.

Individual colonies were expanded and maintained on mouse embryonic feeder cells in ES cell medium, comprising DMEM (D6546; Sigma-Aldrich), 15% FCS (ES-009; Sigma-Aldrich), 100 units/ml penicillin, 100 µg/ml streptomycin, 1 mM sodium pyruvate, 1 mM L-glutamine, 0.15 mM monothioglycerol, 1× nonessential amino

acids, and $10^3$ U/ml leukaemia inhibitory factor (ESGRO; Millipore) after 7 d of 300 $\mu$g/ml neomycin selection.

## ESC differentiation

ESCs were differentiated as previously described (Gilmour et al, 2014; Regha et al, 2015) with the following modifications. FLK1+ cells were purified by magnetic cell sorting, using biotin-conjugated CD309 antibody (eBioscience), anti-biotin microbeads (Miltenyi Biotec), and LS columns (Miltenyi Biotec) after culture of embryoid bodies for between 3.25 and 3.75 d (cell line dependent, when 30–60% of cells were FLK1+, 3.25 for RUNX1-EVI1 line, 3.5 for RUNX1-ETO line, 3.75 d for R201Q, and R204X lines). These FLK1+ cells were then cultured in gelatin-coated flasks – 1.2–1.4 × $10^6$ cells in a T150 flask to form the blast culture. After 1–2 d (cell line dependent, when blast culture comprised ~30% HE, ~40% progenitors, 1 d for RUNX1-EVI1 line, 2 d for R201Q, R204X, and RUNX1-ETO lines), 0.1–0.5 $\mu$g/ml doxycycline was added as appropriate and cells were cultured in the same media for a further 18 h before sorting for HE and progenitors.

## Flow cytometry

Cell populations were identified and sorted on day 2–3 of blast culture based on surface markers. Cells were stained with cKit-APC (BD pharmingen), Tie2-PE (eBioscience), and CD41-PE-Cy7 (eBioscience), or CD41-PE-Cy7 and CD11b-PE (eBioscience) and analysed on a Cyan ADP flow cytometer (Beckman Coulter) with data analysis using FlowJo, or sorted on a FACS Aria cell sorter (BD Biosciences).

## CFU assays

Unsorted floating cells were taken from d2 to 3 of the blast culture, and 5 × $10^3$ cells were seeded in 1 ml MethoCult (M3434; STEMCELL Technologies) per dish, in duplicate and counted after 10 d. For flow cytometry analysis, MethoCult containing cells was harvested in ice cold MACS buffer, and duplicates were pooled and washed with MACS buffer before staining to remove MethoCult.

## Western blotting

20 $\mu$g of protein extracts in Laemmli buffer were run on a 4–20% gradient pre-cast gel (Bio-Rad) and transferred to nitrocellulose using Turbo transfer packs (Bio-Rad). Membranes were blocked using 5% milk in TBS-T, then RUNX1 (C-terminal: ab23980, 1:3,000; Abcam or N-terminal: sc-8563 N-20, 1:250; Santa Cruz Biotechnology) or anti-HA (H6908, 1:1,000; Sigma-Aldrich) was applied overnight at 4°C in 5% milk in TBS-T. After washing in TBS-T, this was followed with HRP-conjugated anti-rabbit or anti-goat antibody (Cell Signalling Technologies), and enhanced chemiluminescent reagent (Amersham) was applied and the blot was visualised using a GelDoc system (Bio-Rad). For loading controls, the membranes were stripped using Restore Stripping Buffer (Thermo Fisher Scientific) and GAPDH (ab8245; Abcam) was applied and visualised as above.

## RNA-seq

RNA was isolated from sorted cells using either the NucleoSpin RNA Kit (Macherey-Nagel) or TRIzol reagent (Thermo Fisher Scientific). RNA-seq libraries were prepared from two biological replicates using the True-Seq Stranded Total RNA Kit (Illumina) and sequenced paired-end in a pool of 12 indexed libraries using a Next-Seq 500/550 High Output Kit v2 150 cycles (Illumina) at the Genomics Birmingham sequencing facility.

## ATAC-seq

ATAC-seq was performed essentially as described (Buenrostro et al, 2015), briefly, 50,000 cKit+CD41+Tie2-progenitors were sorted by FACS and transposed in 1× Tagment DNA buffer (Illumina), Tn5 transposase (Illumina), and 0.01% Digitonin (Promega) for 30 min at 37°C with agitation. For R204X, RUNX1-ETO, and RUNX1-EVI1 samples the tagmentation buffer additionally contained 0.3× PBS and 0.1% Tween-20. DNA was purified using a MinElute Reaction Cleanup Kit (QIAGEN). DNA was amplified by PCR using Nextera primers and libraries were sequenced using a Next-Seq 500/550 High Output Kit v2 75 cycles (Illumina).

## ChIP-seq

ChIP was performed as previously described (Obier et al, 2016; Kellaway et al, 2020) with the following modifications. cKit+ progenitors were sorted by MACS, and for PU.1 and H3K27ac cross-linked only in 1% formaldehyde (single crosslinking), or with both 415 $\mu$g/ml DSG, followed by formaldehyde (double cross-linking) for RUNX1 and GATA1. For single cross-linked cells, nuclei were sonicated for four cycles of 30 s on/30 s off using a Picoruptor (Diagenode). Immunoprecipitation was carried out overnight at 4°C using 2 $\mu$g of RUNX1 antibody (C-terminal, ab23980; Abcam), PU.1 antibody (sc-352; Santa Cruz Biotechnology) or GATA1 antibody (ab11852; Abcam), or for 4 h at 4°C using 1 $\mu$g of H3K27ac antibody (ab4729; Abcam) coupled to 15 $\mu$g Dynabeads Protein G (Invitrogen) per 2 × $10^6$ cells. DNA from 2 to 3 immunoprecipitations was pooled for RUNX1, but just one immunoprecipitation for H3K27ac, and extracted using Ampure beads (Beckman Coulter). ChIP libraries were generated using the KAPA HyperPrep Kit, libraries were size-selected to obtain fragments between 150 and 450 bp, and were sequenced as for ATAC-seq.

## Immunocytochemistry

5 × $10^5$ cells were adhered to microscope slides using a Cytospin cytocentrifuge (Thermo Fisher Scientific) for 3 min at 200$g$ and fixed in 4% formaldehyde (Pierce) for 15 min. Cells were permeabilised in 0.1% Triton X-100 and nonspecific staining was prevented by incubation in 3% bovine serum albumin. Antibodies were applied for 1 h at room temperature before washing, anti-HA (H6908; Sigma-Aldrich) at 1:200, anti-EVI1 (2593; Cell Signalling Technology) at 1:200, anti-RUNX1 (C-terminal, sc-28679 H-65; Santa Cruz Biotechnology) at 1:200 or anti-CBF$\beta$ (sc-56751; Santa Cruz Biotechnology), and secondary Alexa Fluor 488–conjugated anti-rabbit (Jackson ImmunoResearch) at 1:200. Slides were mounted with ProLong

Gold antifade reagent with DAPI (Invitrogen). Slides were visualised using a Zeiss LSM 780 equipped with a Quasar spectral (GaAsP) detection system, using a Plan Achromat 40× 1.2 NA water immersion objective, Lasos 30 mW Diode 405 nm, Lasos 25 mW LGN30001 Argon 488, and Lasos 2 mW HeNe 594 nm laser lines. Images were acquired using Zen black version 2.1. Post-acquisition brightness and contrast adjustment was performed uniformly across the entire image.

### PLA

Cells were prepared, fixed, and blocked as for immunocytochemistry. Primary antibodies (sources as for immunocytochemistry) were applied in pairs—anti-CBFβ at 1:100, with either anti-RUNX1 at 1:20, anti-HA at 1:250 or anti-EVI1 at 1:100 for 1 h at room temperature. Probes, ligation, and amplification solutions (Duolink; Sigma-Aldrich) were then applied at 37°C according to the manufacturer's instructions, and the slides were mounted in Duolink mounting medium with DAPI (Sigma-Aldrich). The slides were visualised as for immunocytochemistry. Post-acquisition brightness and contrast adjustment was performed uniformly across the entire image.

### RNA-seq analysis

Raw paired-end reads were processed to remove low quality sequences with Trimmomatic v0.38 (Bolger et al, 2014). Processed reads were then aligned to the mouse genome (mm10) using Hisat2 v2.1.0 (Kim et al, 2015) with default parameters. Read counts were calculated using featureCounts v1.5.1 (Liao et al, 2013) with the options -p -s 2. Gene models from refSeq (O'Leary et al, 2015) were used as the reference transcriptome. Only genes that were detected with at least 50 reads in at least one sample were retained for further analysis. Differential gene expression analysis was carried out using the voom method (Law et al, 2014) in the limma package v3.40.6 (Ritchie et al, 2015) in R v3.6.1. A gene was considered to be differentially expressed if it had a fold change of at least two and an adjusted $P$-value less than 0.05.

Clustering of gene expression data was carried out by first calculating pairwise Pearson correlations of the $\log_2$-transformed fold changes for each pair of samples in R, and these were then hierarchically clustered using complete linkage of the Euclidean distances and plotted as a heat map in R.

### ATAC-seq analysis

Single-end reads from ATAC-seq experiments were processed with Trimmomatic v0.38. Reads were then aligned to the mouse genome (mm10) with Bowtie2 v2.2.3 (Langmead & Salzberg, 2012) using the options –very-sensitive-local. Potential PCR duplicates were removed from the alignments using the MarkDuplicates function in Picard v2.10.5 (http://broadinstitute.github.io/picard). Regions of open chromatin (peaks) were identified using MACS2 v2.1.1 (Zhang et al, 2008) using the options -B –trackline –nomodel. The resulting peaks were then filtered against the mm10 blacklist (Amemiya et al, 2019) to remove potential artefacts from the data. Peaks were then annotated as either promoter-proximal if within 1.5 kb of a transcription start site, and as a distal element otherwise. Promoter-proximal and distal elements were treated separately in all further analysis.

To carry out differential chromatin accessibility analysis, a peak union was first constructed by merging peaks from the –dox and +dox samples that had summits within 400 bp of each other using the merge function in bedtools v2.26.0 (Quinlan & Hall, 2010). A new summit position was then defined for these peaks as the mid-point between the original summits. The average tag-density in a 400-bp window centred on the peak summits was retrieved from the bedGraph files produced by MACS2 during the peak calling step. This was done using the annotatePeaks.pl function in Homer v4.9.1 (Heinz et al, 2010) with the options -size 400 -bedGraph. These were then normalised as counts-per-million (CPM) in R v3.6.1 and further $\log_2$-transformed as $\log_2$(CPM + 1). A peak was considered to be differentially accessible if it had at least a twofold difference between –dox and +dox conditions. A de-novo motif analysis was carried out in the sets of gained and lost peaks using the findMotifsGenome.pl function in Homer using the options -size 200 -noknown.

Hierarchical clustering of ATAC-seq data was carried out using the $\log_2$-transformed normalised tag-counts. Pairwise Pearson correlation values were calculated for each pair of samples and clustered using complete linkage of the Euclidean distances and plotted as a heat map in R.

Tag density plots were constructed by retrieving the tag-density in a 2 kb window centred on the peak summits with the annotatePeaks.pl function in Homer with the options -size 2000 -hist 10 -ghist -bedGraph. These were then plotted as a heat map using Java TreeView v1.1.6 (Saldanha, 2004).

### ChIP-seq analysis

RUNX1-ETO, RUNX1-EVI1, and the RUNX1 ChIP-seq datasets from the RUNX1-ETO– and RUNX1-EVI1–expressing cells (Regha et al, 2015; Kellaway et al, 2020) were downloaded from the Gene Expression Omnibus (GEO) under accession numbers GSE64625 and GSE143460.

Sequencing reads from ChIP-seq experiments were processed, aligned, and de-duplicated as described above for the ATAC-seq data. Peaks from the RUNX1, RUNX1-ETO, RUNX1-EVI1, GATA1, and PU.1 ChIP-seq data were called using MACS2 v2.6.1 with the options –keep-dup all -B –trackline -q 0.01. Peaks from the H3K27ac ChIP-seq data were also called using MACS2, but with addition of the –broad option. Only peaks that were found within open chromatin, as measured by the ATAC-seq data were retained for further analysis. Differential peak analysis was carried out in the same way as the differential chromatin accessibility analysis described above for the ATAC-seq data with a modification for the H3K27ac data for which the window to calculate the tag density was increased to 800 bp to count reads which flank the open chromatin. To identify potential targets for each of the transcription factors measured, we annotated the peaks to their closest gene using the annotatePeaks.pl function in homer v4.9.1. Average profiles were constructed from the ChIP-seq data using deepTools v3.3.2 (Ramírez et al, 2016). To do this, read counts were calculated and normalised as CPM using the bamCoverage function in deepTools, the average profile calculated using the computeMatrix function with the reference-point option, and then plotted in R. Shared sites were calculated

using bedtools intersect and plotted using the UpSetR function in R. Tag-density plots were constructed as described above for the ATAC-seq data.

## Motif enrichment analysis

To identify transcription factor binding–motifs that are enriched in a set of peaks relative to another, we calculated a motif enrichment score, $S_{ij}$ for each motif i in each peak set j as:

$$S_{ij} = \frac{n_{ij}/m_j}{\sum_j n_{ij} / \sum_j m_j},$$

where $n_{ij}$ is the number of sites in peak set j that contain the motif i and $m_j$ is the total number of sites in peak set j. This was calculated for each TF motif in each of the peak sets being considered and produced a matrix of enrichment scores which were then hierarchically clustered using complete linkage of the Euclidean distance in R and displayed as a heat map. The set of motif probability weight matrices used for this analysis were derived from a de-novo motif search of the gained and lost ATAC-seq peaks using Homer.

Motif density plots were constructed by retrieving the motif density in a 2 kb window centred on the peak summits with the annotatePeaks.pl function in homer with the options -size 2000 -hist 10 -ghist –m, using the Homer known motif database. These were then plotted as a heat map using Java TreeView v1.1.6.

## Lineage priming analysis

To determine if the sets of +dox- and –dox-specific ATAC peaks that we found may also contain a chromatin signature that is normally only associated with a particular cell type, we carried out an analysis designed to measure if the number of cell type–specific sites that are also found in our ATAC-seq data is significantly different from what would be expected by chance.

To do this, we downloaded a set of ATAC-seq data that were generated from a number of mature cells types by (Lara-Astiaso et al, 2014; Heuston et al, 2018). These data were downloaded from the GEO under accession numbers GSE59992 and GSE143270. The cell types considered here were CMPs, B-cells, monocytes, erythroblasts, and megakaryocytes. These ATAC-seq data were aligned and peaks were called and filtered as described above. Only peaks that were found in both replicates for each cell type were retained for further analysis. A peak was then considered to be cell type specific if it was found in only one of the cell types. This was done by comparing the peaks from each cell type to the union of peaks from all other cell types using the intersect function in bedtools with the -v parameter.

To determine if any of these cell type–specific peak sets were either significantly enriched or depleted in our data, we carried out a randomisation-based test for each of our RUNX1 mutant +dox- and –dox-specific peak sets as follows. First, we counted the number of +dox-specific peaks that overlap with the cell type–specific peaks using the intersect function in pybedtools (Dale et al, 2011). We then randomly sampled a set of peaks from the full set of distal sites in that RUNX1 mutant, and counted the number of

overlapping peaks between this random set and the cell type–specific peaks. The number of peaks sampled was equal to that of the +dox peaks and could be sampled anywhere from the +dox, –dox, and shared peaks. This procedure was repeated 1,000 times and produced a list of counts measuring the overlap of the random sets with the cell type–specific peaks. These counts were then used to calculate a Z-score using the formula:

$$z = \frac{x - \mu}{\sigma},$$

where x is the number of +dox peaks that overlap a cell type–specific peak, $\mu$ is the mean of the counts from the re-sampling procedure, and $\sigma$ is the SD of those counts. A P-value measuring the statistical significance of the enrichment could also be derived from this test. This was calculated as the proportion of times the number of overlapping sites from the random peak sets was greater than that of the actual +dox peaks, with a low P-value suggesting that the number of cell type specific peaks found in the +dox peaks is greater than what would be expected only by chance. A P-value measuring depletion could also be calculated and here is calculated as the proportion of times the number of overlapping sites from the random peak sets was less than that of the actual +dox peaks. In this case, a low P-value suggests that the cell type specific peaks are under-represented in the +dox specific peaks. This same test was also applied to each of the –dox-specific peak sets.

# Data Availability

All sequencing data from this publication have been deposited to GEO and assigned the identifier GSE154623.

# Supplementary Information

# Acknowledgements

This work was funded by grants from the Kay Kendall Leukaemia Fund, the Biotechnology and Biological Sciences Research Council, and Blood Cancer UK (Bloodwise) to C Bonifer. We thank Genomics Birmingham for their expert sequencing service, the University of Birmingham Flow Cytometry unit for cell sorting, and Martin Higgs from the Institute of Cancer and Genomic Sciences for help with the PLA assay.

## Author Contributions

SG Kellaway: formal analysis, investigation, visualization, methodology, and writing—original draft, review, and editing.
P Keane: formal analysis, visualization, and writing—review and editing.
B Edginton-White: formal analysis and writing—review and editing.
K Regha investigation.
E Kennett: investigation.

C Bonifer: conceptualization, supervision, funding acquisition, and writing—original draft, review, and editing.

## Conflict of Interest Statement

The authors declare that they have no conflict of interest.

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
