## [Reviewer comments · Life Science Alliance]

Life Science Alliance

Different mutant RUNX1 oncoproteins program alternate haematopoietic differentiation trajectories

Sophie Kellaway, Peter Keane, Benjamin Edginton-White, Regha Kakkad, Ella Kennett, and Constanze Bonifer

DOI: <https://doi.org/10.26508/lsa.202000864>

Corresponding author(s): Constanze Bonifer, University of Birmingham and Sophie Kellaway, University of Birmingham

Review Timeline:

Submission Date:	2020-07-29
Editorial Decision:	2020-09-25
Revision Received:	2020-11-24
Editorial Decision:	2020-12-01
Revision Received:	2020-12-03
Accepted:	2020-12-07

Scientific Editor: Shachi Bhatt

Transaction Report:

September 25, 2020

Re: Life Science Alliance manuscript #LSA-2020-00864-T

Prof. Constanze Bonifer
University of Birmingham
Institute for Cancer and Genomic Sciences
Institute for Biomedical Research
Birmingham, West Midlands B15 2TT
United Kingdom

Dear Dr. Bonifer,

Thank you for submitting your manuscript entitled "Different mutant RUNX1 oncoproteins program alternate haematopoietic differentiation trajectories" to Life Science Alliance. The manuscript was assessed by expert reviewers, whose comments are appended to this letter. As you will note from the comments below, the referees are quite enthusiastic about the study, and have raised minor addressable concerns for revision. We would like to invite you to submit a revised version to Life Science Alliance that addresses all the referees' concerns.

We would be happy to discuss the individual revision points further with you should this be helpful. While you are revising your manuscript, please also attend to the below editorial points to help expedite the publication of your manuscript. Please direct any editorial questions to the journal office.

The typical timeframe for revisions is three months. Please note that papers are generally considered through only one revision cycle, so strong support from the referees on the revised version is needed for acceptance. When submitting the revision, please include a letter addressing the reviewers' comments point by point.

Thank you for considering Life Science Alliance as an appropriate venue for your research. We look forward to receiving your revised manuscript.

Sincerely,

Shachi Bhatt, Ph.D.
Executive Editor
Life Science Alliance

- A letter addressing the reviewers' comments point by point.
- An editable version of the final text (.DOC or .DOCX) is needed for copyediting (no PDFs).
- High-resolution figure, supplementary figure and video files uploaded as individual files: See our detailed guidelines for preparing your production-ready images, <https://www.life-science-alliance.org/authors>
- Summary blurb (enter in submission system): A short text summarizing in a single sentence the study (max. 200 characters including spaces). This text is used in conjunction with the titles of papers, hence should be informative and complementary to the title and running title. It should describe the context and significance of the findings for a general readership; it should be written in the present tense and refer to the work in the third person. Author names should not be mentioned.

B. MANUSCRIPT ORGANIZATION AND FORMATTING:

Reviewer #1 (Comments to the Authors (Required)):

This study examines the chromatin, expression, regulatory network changes in the ESC model of hematopoietic differentiation when the expression of mutant or fusion Runx1 proteins associated with AML are induced. Highly specialized single cell methods examining the DNA binding of the four mutant Runx1 proteins along with ATACseq, RNAseq and ChIP are used. This lab has championed the use of these difficult analyses in the mouse ESC hematopoietic differentiation to provide important new data on how Runx1 is pivotal to hematopoietic development. The data are clearly presented and statistical significance is shown for the datasets. The conclusion that each of the mutants lead to different associated outcomes are well grounded in the data and are important new information in the field of gene regulation, developmental and malignant hematopoiesis.

Minor comment

While the manuscript is well-written the use of the word 'cause' in the abstract and introduction to describe the role of Runx1 in AML is too strong.....Runx1 mutations and fusions are 'associated' with AML.

The sentence on page 6, lines 5 to 9, is difficult to understand. It is unclear what the authors are trying to state.

Reviewer #2 (Comments to the Authors (Required)):

The manuscript by Kellaway and colleagues explores how different RUNX1 mutants affect chromatin accessibility, histone acetylation and gene expression during hematopoietic differentiation of mouse ESCs. RUNX1 is frequently mutated in human leukemia and how different mutations affect hematopoietic differentiation is of interest to the field. The authors are experts in multi-omics analyses and have used in vitro differentiation models successfully in previous studies. There are however some concerns that need to be addressed prior to publication. As there are no page numbers, these are as much as possible listed in the order of appearing in the paper.

1. Intro/Results: The rationale for picking specifically R201Q (R174Q) and R204X (R177X) could be clarified further, as there are different DBD and TAD mutants described in the literature.
2. Induction of mutant gene expression: it is stated that expression levels of the transgenes were 'approximately equal' to endogenous Runx1. Suppl.Fig.1A suggests otherwise. This should be clarified. This would help assessing the results. What are the double bands in the Western blot at the right side of the figure?
3. Could the specificity of the antibodies used be stated more explicitly. E.g. in a schematic? This relates not only to the WB, but also the ChIP-Seq and PLA assays which without this explicit information are a little complex to interpret for the reader.
4. The authors state "As differentiation in this system is not entirely synchronous, timing of induction was adjusted in a cell line specific manner such that it occurred in approximately the same target cell populations ensuring that results were comparable (Supplementary Figure 1B)." However, it is not clear how the timings differed and were monitored, which makes it impossible for anyone to try to replicate the study. No data are provided in support of the statement that dox treatment was started at the "onset of the RUNX1 transcriptional program".
5. Suppl.Fig.1D: In the absence of any labeling, it is not clear how these images support the statement that fewer megakaryocytes are observed following induction of R201Q in the mixed lineage colonies. Could this statement be supported by quantitative data obtained by flow or CFU-Meg? Particularly since this links to the results shown in Figure 7 (see also comment 13 below).
6. Figure 2B and Suppl.Fig.2D: Please correct R203X to R204X.
7. "We found that changes to chromatin accessibility and gene expression were largely driven by mutant-specific changes in the endogenous RUNX1 binding patterns (Figure 2)" is not backed up by experimental evidence. It may be more appropriate to use "associated with" instead of "driven by".
8. PLA in Figure 3: The conclusion of this paragraph is "Displacement of endogenous RUNX1 binding by the mutant RUNX1 proteins was only found in the case of the two fusion proteins." However, earlier in the paragraph it says " RUNX1-ETO and R204X expression caused no change to the quantity of RUNX1/CBF β interactions, suggesting the changes in RUNX1 binding were due to displacement, .." These statements seem contradictory. It would be good to have the data for HA-mutant/CBF β PLA for all 4 mutants.

9. RNA-Seq: Reference to the supplementary table (DE gene lists w/wo Dox) could be clearer. Is there a way in which to make the description of the pairwise comparisons (Figure 4 B,C) more visual and easier to follow? Similarly, Fig 4D is not very intuitive - is there another way to show this?
10. "Next we analysed which of the genes with altered expression were direct targets of either RUNX1 or the two fusion proteins." This needs to be clarified. Presumably this was done based on RUNX1 ChIP-Seq from a previous study?
11. Concluding sentence of text accompanying Fig 4: "... R204X were RUNX1 targets but binding of RUNX1 was unchanged again indicating that this oncoprotein perturbs the action of RUNX1 at its binding sites rather than disrupting binding itself." Is displacement meant here (as stated earlier)?
12. "RUNX1-EVI1, PU.1 binding was maintained but was more prevalent at those sites where chromatin accessibility was gained." There seems to be an overall decrease in PU.1 binding in the Runx1-EVI1 ChIP?
13. Please add the cell type in the legend of Fig 5A, B, E.
14. Based on the transcription profiling results (Fig 7) predictions are made as to the effect the different mutations have on hematopoietic differentiation. Earlier in the paper, the authors mentioned that they saw less megakaryocytes upon R201Q induction (supplementary 1D) and this matches with the lost accessible chromatin sites for megakaryocyte in Fig 7B. For another example, Fig 7B shows that erythroblast associated sites are lost upon Runx1-EVI1 induction and the number of erythroid colonies decrease upon Runx1-EVI1 induction for primary CFU-C assay in 1D. It would be useful to perform functional assays like CFU-Cs, flow cytometry etc to validate the predictions from Figure 7B for the other mutations. At the least available data could be compared more directly to functional output (this paper or literature) in the discussion.

Dear Editor,

We thank all reviewers for their constructive suggestions. In response, we have now conducted more experiments and analyses to address the vast majority of the referees' comments. All changes are marked in red. Our response to the comments is as follows:

Reviewer #1 (Comments to the Authors (Required)):

This study examines the chromatin, expression, regulatory network changes in the ESC model of hematopoietic differentiation when the expression of mutant or fusion Runx1 proteins associated with AML are induced. Highly specialized single cell methods examining the DNA binding of the four mutant Runx1 proteins along with ATACseq, RNAseq and ChIP are used. This lab has championed the use of these difficult analyses in the mouse ESC hematopoietic differentiation to provide important new data on how Runx1 is pivotal to hematopoietic development. The data are clearly presented and statistical significance is shown for the datasets. The conclusion that each of the mutants lead to different associated outcomes are well grounded in the data and are important new information in the field of gene regulation, developmental and malignant hematopoiesis.

Response: We thank the reviewer for the positive evaluation of our study

Minor comment

While the manuscript is well-written the use of the word 'cause' in the abstract and introduction to describe the role of Runx1 in AML is too strong.....Runx1 mutations and fusions are 'associated' with AML.

Response: This has been changed as suggested

The sentence on page 6, lines 5 to 9, is difficult to understand. It is unclear what the authors are trying to state.

Response: We have rewritten this section to clarify that 1. Reduction of endogenous RUNX1 binding in ChIP was found reproducibly and 2. We were also unable to detect R201Q binding via ChIP.

Reviewer #2 (Comments to the Authors (Required)):

The manuscript by Kellaway and colleagues explores how different RUNX1 mutants affect chromatin accessibility, histone acetylation and gene expression during hematopoietic differentiation of mouse ESCs. RUNX1 is frequently mutated in human leukemia and how different mutations affect hematopoietic differentiation is of interest to the field. The authors are experts in multi-omics analyses and have used in vitro differentiation models successfully in previous studies. There are however some concerns that need to be addressed prior to publication. As there are no page numbers, these are as much as possible listed in the order of appearing in the paper.

Response: We thank the reviewer for their thorough review which has helped to improve the clarity of a complex and challenging study, and strengthened the conclusions detailed within. We apologize for making their life difficult by omitting page numbers.

1. Intro/Results: The rationale for picking specifically R201Q (R174Q) and R204X (R177X) could be clarified further, as there are different DBD and TAD mutants described in the literature.

Response: We have added a sentence to the introduction and the results to explain the reason. As the reviewer rightly says, many RUNX1 mutations have been described in the literature and so we chose to use 2 of those which were studied in both Matheny et al and Michaud et al to allow comparison with these studies.

2. Induction of mutant gene expression: it is stated that expression levels of the transgenes were 'approximately equal' to endogenous Runx1. Suppl.Fig.1A suggests otherwise. This should be clarified.

This would help assessing the results. What are the double bands in the Western blot at the right side of the figure?

Response:

Note that the antibody used to assess R201Q induction was raised against the C-terminus of the human protein and shows only weak cross-reaction with the endogenous mouse protein as here the C-terminal epitope differs in four amino acids, which explains the weak signal in the -dox sample. The N-terminal epitopes are identical between human and mouse. Therefore, the difference between the signals in -dox and +dox lanes are an average 7 fold instead of 2-3 fold (twice repeated). This fact is now mentioned in the figure legend. To further substantiate our statement, qPCR data has been added. The text has been revised to indicate that the induced RUNX1 protein are expressed at levels of the endogenous protein rather than exactly the same or extensive overexpression. The additional band present with the N-terminal RUNX1 antibody most likely represents non-specific binding or post-translational modification, we do not know which but this does not change interpretation of the blot as it is present in both conditions at the same level – this information has been added to the figure legend.

3. Could the specificity of the antibodies used be stated more explicitly. E.g. in a schematic? This relates not only to the WB, but also the ChIP-Seq and PLA assays which without this explicit information are a little complex to interpret for the reader.

Response: Reactivity of the antibodies is now indicated in the schematic in Figure 1. The text/figures have been modified throughout to indicate when C/N-terminal RUNX1 antibodies were used.

4. The authors state "As differentiation in this system is not entirely synchronous, timing of induction was adjusted in a cell line specific manner such that it occurred in approximately the same target cell populations ensuring that results were comparable (Supplementary Figure 1B)." However, it is not clear how the timings differed and were monitored, which makes it impossible for anyone to try to replicate the study. No data are provided in support of the statement that dox treatment was started at the "onset of the RUNX1 transcriptional program".

Response: Precise details on the timings used and the populations these were based on have been added to the methods, and details on the induction population composition added to the main text. However, were this to be replicated these timings may vary again with new cell lines. The onset of RUNX1 transcription in the EHT is a gradient so induction was during this process when there are cells present both with and without the RUNX1 transcriptional program, as indicated by the newly added population composition, we have also added a citation for this statement.

5. Suppl.Fig.1D: In the absence of any labeling, it is not clear how these images support the statement that fewer megakaryocytes are observed following induction of R201Q in the mixed lineage colonies. Could this statement be supported by quantitative data obtained by flow or CFU-Meg? Particularly since this links to the results shown in Figure 7 (see also comment 13 below).

Response: We have added arrows to the images to indicate the megakaryocytes and performed flow cytometry on these same cells to quantify the changes as suggested, a reduction was also seen in the flow cytometry data.

6. Figure 2B and Suppl.Fig.2D: Please correct R203X to R204X.

Response: We apologise for this typo, it has now been corrected.

7. "We found that changes to chromatin accessibility and gene expression were largely driven by mutant-specific changes in the endogenous RUNX1 binding patterns (Figure 2)" is not backed up by experimental evidence. It may be more appropriate to use "associated with" instead of "driven by".

Response: We have changed this as suggested.

8. PLA in Figure 3: The conclusion of this paragraph is "Displacement of endogenous RUNX1 binding by the mutant RUNX1 proteins was only found in the case of the two fusion proteins." However, earlier in the paragraph it says "RUNX1-ETO and R204X expression caused no change to the quantity of RUNX1/CBF β interactions, suggesting the changes in RUNX1 binding were due to displacement, .." These statements seem contradictory. It would be good to have the data for HA-mutant/CBF β PLA for all 4 mutants.

Response: This was poorly phrased, we have rewritten this to be clear it was only referring to RUNX1-ETO causing displacement of RUNX1, not R204X. HA-mutant/CBF β PLA is present for the three mutants with HA tags, and the EVI1 antibody was used instead for RUNX1-EVI1. This is hopefully clearer now with the antibody schematic in Figure 1 and has also been explicitly referred to in the text to aid clarity.

9. RNA-Seq: Reference to the supplementary table (DE gene lists w/wo Dox) could be clearer. Is there a way in which to make the description of the pairwise comparisons (Figure 4 B,C) more visual and easier to follow? Similarly, Fig 4D is not very intuitive - is there another way to show this?

Response: We have added a reference to the supplementary table in the text. We appreciate Figure 4B contains a lot of data which makes it difficult to interpret, we have added an explicit description in the text and further colouring to the figure which we hope helps. We have split Figure 4D so up and downregulated genes are separate and improved the legend, hopefully this is also easier to understand now.

10. "Next we analysed which of the genes with altered expression were direct targets of either RUNX1 or the two fusion proteins." This needs to be clarified. Presumably this was done based on RUNX1 ChIP-Seq from a previous study?

Response: This was based on the RUNX1 ChIP-seq data generated in this paper (Figure 2), allowing us to also see whether the genes were deregulated in response to changing RUNX1 binding. This has been expanded upon in the text to clarify

11. Concluding sentence of text accompanying Fig 4: "... R204X were RUNX1 targets but binding of RUNX1 was unchanged again indicating that this oncoprotein perturbs the action of RUNX1 at its binding sites rather than disrupting binding itself." Is displacement meant here (as stated earlier)?

Response: We observed no displacement of RUNX1 binding due to R204X so this statement is correct, and in light of the changes in response to point 8 this should now be clearer.

12. "RUNX1-EVI1, PU.1 binding was maintained but was more prevalent at those sites where chromatin accessibility was gained." There seems to be an overall decrease in PU.1 binding in the Runx1-EVI1 ChIP?

Response: Thank you for spotting this, further analysis has confirmed that this is correct. We have added an average profile to Supplementary Figure 5 to show this and changed the text accordingly.

13. Please add the cell type in the legend of Fig 5A, B, E.

Response: This has been done.

14. Based on the transcription profiling results (Fig 7) predictions are made as to the effect the different mutations have on hematopoietic differentiation. Earlier in the paper, the authors mentioned that they saw less megakaryocytes upon R201Q induction (supplementary 1D) and this matches with the lost accessible chromatin sites for megakaryocyte in Fig 7B. For another example, Fig 7B shows that erythroblast associated sites are lost upon Runx1-EVI1 induction and the number of erythroid colonies decrease upon Runx1-EVI1 induction for primary CFU-C assay in 1D. It would be useful to perform functional assays like CFU-Cs, flow cytometry etc to validate the predictions from Figure 7B for the other mutations. At the least available data could be compared more directly to functional output (this paper or literature) in the discussion.

Response: The reviewer is correct that many of the predictions based on chromatin accessibility are reflected by the phenotypic data, this is further strengthened by the experiment carried out for point 5. We have added further text to the discussion to highlight those predictions which matched up to the CFU-C assays as well as noting those which could not be validated in this system.

In summary, we thank the reviewers for their careful review of the paper which made it much better and we hope that it is now suitable to be published in Life Science Alliance.

Yours sincerely

Sophie Kellaway and Constanze Bonifer

December 1, 2020

RE: Life Science Alliance Manuscript #LSA-2020-00864-TR

Prof. Constanze Bonifer
University of Birmingham
Institute for Cancer and Genomic Sciences
Institute for Biomedical Research
Birmingham, West Midlands B15 2TT
United Kingdom

Dear Dr. Bonifer,

Thank you for submitting your revised manuscript entitled "Different mutant RUNX1 oncoproteins program alternate haematopoietic differentiation trajectories". We would be happy to publish your paper in Life Science Alliance pending final revisions necessary to meet our formatting guidelines.

Along with the points listed below please also attend to the following,

- please move the supplemental figure legends from the EV file to the main manuscript
- please provide a legend for the supplemental table (that includes the dataset), and include a call out for the table in the manuscript text
- the ATACseq and other plots have been reused in multiple figures (eg. 2A and 5C, 2A and S6A, and 5C and S5A). While I understand that these might have been repeated for the sake of clarity within each individual figure, we would appreciate if you can clarify so in the respective figure legends

A. FINAL FILES:

B. MANUSCRIPT ORGANIZATION AND FORMATTING:

Sincerely,

Shachi Bhatt, Ph.D.
Executive Editor
Life Science Alliance
<https://www.lsjournal.org/>
Tweet @SciBhatt @LSAJournal

December 7, 2020

RE: Life Science Alliance Manuscript #LSA-2020-00864-TRR

Prof. Constanze Bonifer
University of Birmingham
Institute for Cancer and Genomic Sciences
Institute for Biomedical Research
Birmingham, West Midlands B15 2TT
United Kingdom

Dear Dr. Bonifer,

Thank you for submitting your Research Article entitled "Different mutant RUNX1 oncoproteins program alternate haematopoietic differentiation trajectories". It is a pleasure to let you know that your manuscript is now accepted for publication in Life Science Alliance. Congratulations on this interesting work.

DISTRIBUTION OF MATERIALS:

Again, congratulations on a very nice paper. I hope you found the review process to be constructive and are pleased with how the manuscript was handled editorially. We look forward to future exciting submissions from your lab.

Sincerely,

Shachi Bhatt, Ph.D.

Executive Editor

Life Science Alliance

<https://www.lsjournal.org/>
